# JAK/STAT controls organ size and fate specification by regulating morphogen production and signalling

Carles Recasens-Alvarez[1,*], Ana Ferreira[1,*] & Marco Milán[1,2]

A stable pool of morphogen-producing cells is critical for the development of any organ or tissue. Here we present evidence that JAK/STAT signalling in the *Drosophila* wing promotes the cycling and survival of Hedgehog-producing cells, thereby allowing the stable localization of the nearby BMP/Dpp-organizing centre in the developing wing appendage. We identify the inhibitor of apoptosis dIAP1 and Cyclin A as two critical genes regulated by JAK/STAT and contributing to the growth of the Hedgehog-expressing cell population. We also unravel an early role of JAK/STAT in guaranteeing Wingless-mediated appendage specification, and a later one in restricting the Dpp-organizing activity to the appendage itself. These results unveil a fundamental role of the conserved JAK/STAT pathway in limb specification and growth by regulating morphogen production and signalling, and a function of pro-survival cues and mitogenic signals in the regulation of the pool of morphogen-producing cells in a developing organ.

[1] Institute for Research in Biomedicine (IRB Barcelona), The Barcelona Institute of Science and Technology, Baldiri Reixac, 10-12, 08028 Barcelona, Spain. [2] Institució Catalana de Recerca i Estudis Avançats (ICREA), Pg. Lluis Companys, 23, 08011 Barcelona, Spain. * These authors contributed equally to this work. Correspondence and requests for materials should be addressed to M.M. (email: marco.milan@irbbarcelona.org).

Despite the great differences in size and shape across the animal phyla, the body plan of most organisms is built up by a limited and conserved number of developmental toolkit genes that follow the same principles of animal design. Studies of limb development, both in vertebrates and invertebrates, have been instrumental to our current understanding of the interplay between morphogen function, growth and patterning[1–5]. Morphogens of the BMP/Dpp, Sonic Hedgehog/Hedgehog and Wnt/Wingless families play a conserved role in promoting growth and fate specification within growing limbs, which emerge as outgrowths perpendicular to the major axes of the developing animal. Morphogens are signalling molecules produced and released from a localized source that spread throughout the tissue to form a concentration gradient. These gradients provide a series of concentration thresholds along the tissue that set the transcriptional state of downstream target genes in discrete domains of expression as a function of their distance from the source. These genetic subdivisions are ultimately used to define cell identity and tissue pattern. A complex set of interactions between morphogens and their corresponding signalling pathways contributes to patterning and organizing limb growth along the dorsal–ventral, anterior–posterior and proximal–distal axes.

The developing limbs of *Drosophila* have proved a valuable model system to genetically and molecularly identify morphogens and members of their corresponding signalling pathways. They have also been instrumental to functionally dissect the interplay between morphogens and their biological functions and to unravel the genetic logic of pattern formation. In the wing primordium, Wingless (Wg) plays an instructive role in the specification of the wing appendage[6] and restricts the expression of Vein (Vn), a ligand of the epidermal growth factor receptor (EGFR) involved in the specification of the body wall[7], to the proximal domain. Interestingly, tissue growth contributes to the specification of the wing versus the body wall by modulating the range of Vn and Wg activities[8]. Asymmetric interactions between posterior (P) Hh-producing and anterior (A) Hh-receiving cells induce the expression of Dpp in the centre of the wing field, and Dpp organizes growth and patterning of the developing appendage[9–12]. Symmetric interactions between dorsal (D) and ventral (V) cells, mediated by the complementary and compartment-specific expression of two different ligands of the Notch receptor, induce the expression of Wg at the DV boundary, and Wg organizes the growth and pattern of the developing wing[13–16].

The Unpaired cytokines are interleukin-6-like secreted proteins produced and released from a localized source that spread along the tissue to activate the conserved JAK (Janus Kinase)/STAT (Signal Transducer and Activator of Transcription) signalling pathway[17]. This pathway is involved in the proximal–distal patterning of limb primordia[18–20], and regulates growth and the competitive status of proliferating cells[21,22]. No developmental role of the conserved JAK/STAT pathway has been described so far in vertebrate limbs. Here we present evidence that JAK/STAT is required in a sequential manner in the developing *Drosophila* wing to guarantee the correct fate- and growth-promoting activities of Wg, Hh and Dpp morphogens. Interestingly, JAK/STAT mediates these activities by three distinct mechanisms. Early in development, localized expression of Unpaired and graded activity of the JAK/STAT pathway along the proximal–distal axis restrict the expression of genes regulated by the Vn/EGFR pathway to the body wall, thus ensuring wing fate specification by the activity of Wg. Later in development, JAK/STAT controls organ growth by promoting the survival and cycling of Hh-producing cells, thereby allowing stable expression of the Dpp stripe in the centre of the wing appendage. Finally, the

building of the wing hinge—a cell population that isolates the growing appendage from the surrounding body wall and that is maintained by the activity of JAK/STAT[18,19,23]—contributes to delimiting the organizing activity of Dpp to the growing appendage. Overall, these data reveal a novel role of the JAK/STAT signalling pathway in the control of organ size and fate specification by regulating the production and activity of morphogens and by spatially restricting their organizing and growth-promoting functions. These findings add a new member to the ample repertoire of signalling molecules and corresponding pathways involved in limb development and unveil a role of pro-survival cues and mitogenic signals in limb development.

## Results

**JAK/STAT restricts EGFR to ensure wing specification.** In second instar wing discs, localized expression of Wg and Vn in opposing domains subdivides the primordium into the presumptive wing field and body wall/notum regions, respectively (Fig. 1a,b; refs 6,7). We first monitored at this developmental stage the expression of Unpaired 1 (Upd) using *upd-gal4*, a P-element insertion in the *upd* locus carrying the Gal4 transcriptional activator[24], and the activity of the pathway using the 10xSTAT-GFP reporter[25]. Upd expression was restricted to the most distal domain of the wing disc, in a broader domain than Wg (Fig. 1c and Supplementary Fig. 1), and activation of the pathway was observed throughout the disc, although GFP levels were clearly lower in the most proximal region of the primordium (Fig. 1d and Supplementary Fig. 1). Restricted expression of *upd* to the distal domain of young discs was confirmed by *in situ* hybridization (Supplementary Fig. 1). We next analysed the developmental role of JAK/STAT at this stage of wing development. We used the *scalloped-gal4* (*sd-gal4*) driver, which is expressed at high levels in the entire early wing primordium[8], to drive expression of RNA interference (RNAi) forms against the *Drosophila* Upd receptor (Domeless), JAK kinase (Hop) and STAT transcription factor (STAT92E). Remarkably, the resulting adult wings were either vestigial or absent, and body wall structures were often duplicated (Fig. 1e,f, red arrows, and Supplementary Fig. 1). This phenotype is reminiscent of the *wg* mutant adult phenotype[26]. In the wing disc, expression of the homeodomain protein Homothorax (Hth) is restricted to the presumptive body wall, while the POU homeodomain protein Nubbin (Nub) is expressed in the presumptive wing territory (Fig. 1h,i, refs 27,28). Wg is expressed in the body wall and wing territories of late third instar discs in a characteristic pattern (Fig. 1h). In mature discs in which JAK/STAT signalling had been compromised, Nub was absent or residual in a small group of cells, and the characteristic expression pattern of Hth and Wg in the notum showed a mirror-image duplication (Fig. 1j,k). These results indicate that JAK/STAT is required for proper wing fate specification.

We next checked the expression and activity of Wg and Vn/EGFR in JAK/STAT-depleted wing discs. First, we analysed whether JAK/STAT controls the expression or activity of Wg, or whether it collaborates with the Wg pathway during wing fate specification. While blocking Wg signalling by overexpression of Shaggy/GSK3 in a stripe along the anterior-posterior (AP) compartment induced the cell-autonomous loss of Nub (Fig. 2a, ref. 8), JAK/STAT depletion in the same domain did not have the same effect (Fig. 2b). The early expression of Wg was not affected by JAK/STAT depletion either (Fig. 2c,d). These observations indicate that JAK/STAT does not have an active role in inducing wing fate or in regulating Wg expression. The *Drosophila* Iroquois complex (Iro-C) consists of three genes encoding homeobox transcription factors (*araucan*, *caupolican* and

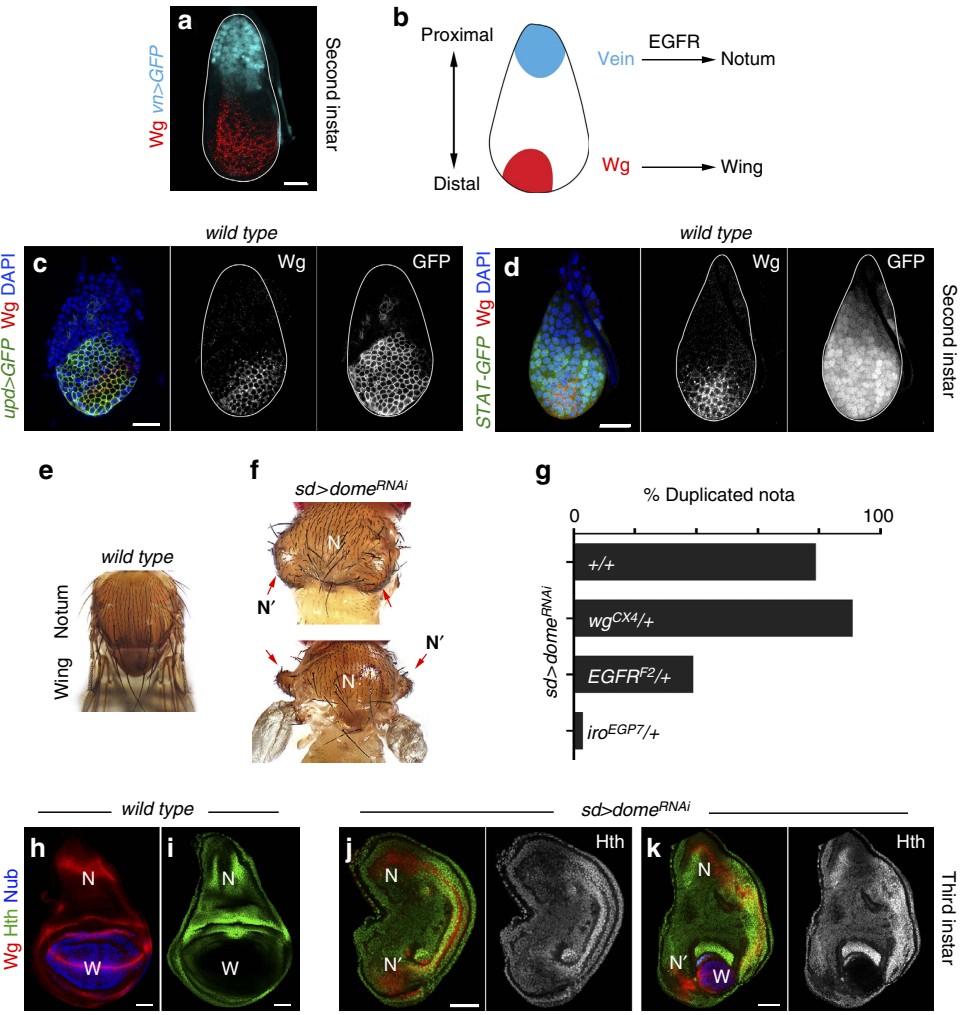

**Figure 1 | Failure to specify wing fate in the absence of JAK/STAT signalling.** (**a,c,d**) Wing primordia of second instar larvae labelled to visualize expression of Wg protein (red or white (**a,c,d**)), *vn* (cyan (**a**), in *vn-gal4, UAS-GFP*), *upd* (green or white (**c**), in *upd-gal4, UAS-myrGFP*), the activity of the JAK/STAT pathway (green or white (**d**), in *10xSTAT-GFP* larvae) and 4,6-diamidino-2-phenylindole (DAPI; blue). The contour of the wing discs is marked by a white line. (**b**) Cartoon depicting the roles of Wg and Vn in the specification of wing and notum territories. (**e,f,h–k**) Adult thoraxes (**e,f**) and mature wing primordia (**h–k**) of *wild-type* male individuals (**e,h,i**) or male individuals expressing *dome^RNAi* under the control of the *sd-gal4* driver (**f,j,k**). Wing primordia were stained for Wg (red, **h,j,k**), Nub (blue, **h,j,k**) and Hth (green or white, **i–k**). Wing territory (W), endogenous nota (N) and duplicated nota territories (N′) are marked. Red arrows in **f** point to the duplicated nota (N′). Adult thoraxes are illustrative examples of complete or partial duplications of the notum structures. Scale bars, 20 μm (**a,c,d**) or 50 μm (**h–k**). (**g**) Histogram plotting the percentage of duplicated nota in the following genotypes: (1) *sd-gal4/Y; UAS-dome^RNAi/+; UAS-dcr2/+* (79.4%, n = 102 heminota). (2) *sd-gal4/Y; UAS-dome^RNAi/wg^CX4; UAS-dcr2/+* (91.5%, n = 106 heminota). (3) *sd-gal4/Y; UAS-dome^RNAi/egfr^F2; UAS-dcr2/+* (39.1%, n = 197 heminota). (4) *sd-gal4/Y; UAS-dome^RNAi/+; iro^EGP7/UAS-dcr2* (2.8%, n = 180 heminota). Only male individuals were scored for each genotype.

*mirror*), which are expressed in the most proximal region of the wing primordium by the activity of Vn/EGFR (Fig. 2c,h) and specify notum structures[29]. Recent experimental evidence has revealed a late role of JAK/STAT in repressing the expression of these transcription factors in the wing hinge of mature primordia[18,19]. We analysed whether JAK/STAT has an earlier and more extensive role in restricting the expression of Vn/EGFR targets such as the *Iro-C* genes and *apterous*, a gene encoding for a homeodomain-containing transcription factor that specifies the D compartment[30]. Interestingly, *mirror* expression was expanded distally in JAK/STAT-depleted second instar primordia (Fig. 2c,d) and this expansion was even more evident in mature wing discs (Fig. 2h–j). In JAK/STAT-depleted primordia, *apterous* expression was also expanded distally (Fig. 2k–m). The expansion in the expression domains of *mirror* and *apterous* was not always accompanied by the loss of the presumptive wing field

(Fig. 2i,l). We found that the initial expression of Vn, which depends on Dpp activity[31], was unaffected in JAK/STAT-depleted early wing primordia (Fig. 2e,f). However, the later expression of Vn, which is reinforced by a positive feedback amplification loop through the activation of the EGFR pathway[7,31], was expanded in JAK/STAT-depleted late second instar (Fig. 2g) and late third instar wing primordia (Fig. 2n–p). The distal expansion in the expression of EGFR target genes contributed to the resulting duplication of notum structures, as halving the doses of the *EGFR* gene or of the whole *iro-C* reduced the frequency of duplicated nota observed in adults (Fig. 1g). This frequency was increased in *wg* heterozygous animals (Fig. 1g).

Interestingly, ectopic expression of Upd to the most proximal side of the wing primordium reduced the expression levels of *mirror* throughout development (Fig. 3a–c), caused a reduction in the size of the notum (Fig. 3d, compared with the inset in Fig. 3d)

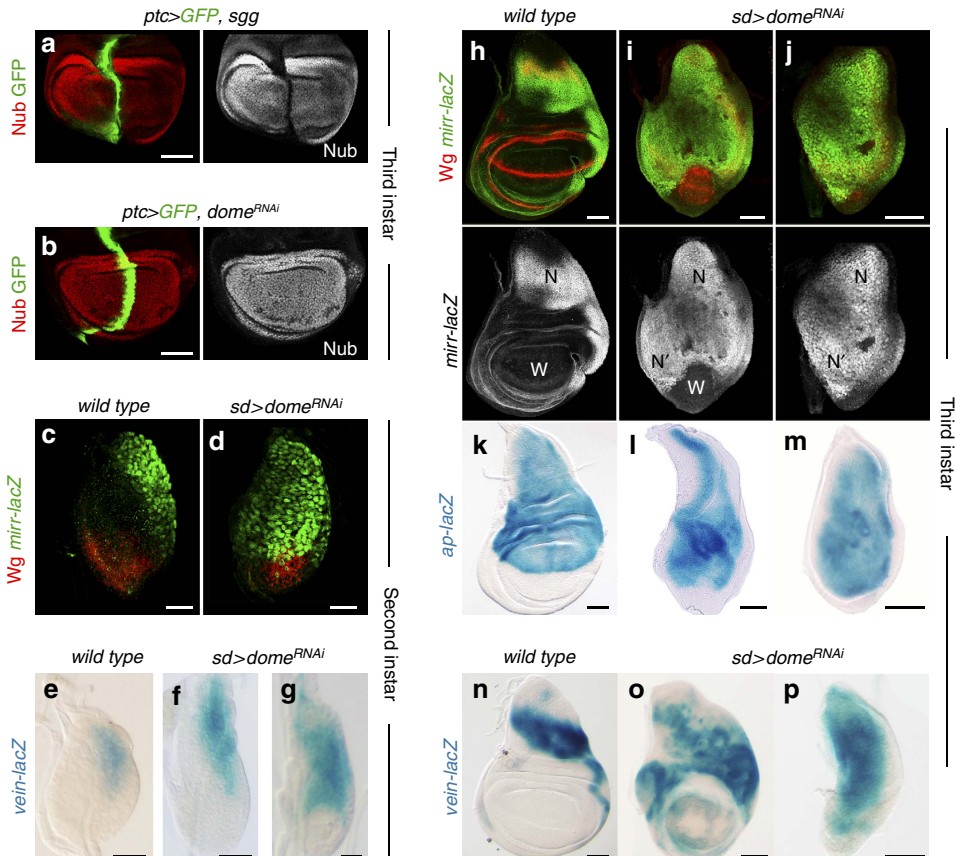

**Figure 2 | JAK/STAT restricts the expression of Vn/EGFR target genes to the body wall.** (**a,b**) Wing primordia of late third instar larvae expressing *shaggy* (*sgg,* **a**) or *dome^{RNAi}* (**b**) under the control of the *ptc-gal4* driver and labelled to visualize Nub protein (red or white) and GFP (green). (**c,d,h–j**) Wing primordia from *wild-type* larvae (**c,h**) or from larvae expressing *dome^{RNAi}* under the control of the *sd-gal4* driver (**d,i,j**) labelled to visualize *mirror* (*mirr-lacZ*, antibody against β-gal, green or white) and Wg protein (red) expression in second (**c,d**) and late third instar (**h–j**) stages. Wing territories (W), endogenous nota (N) and duplicated nota territories (N′) are marked. (**e–g,n–p**) Wing primordia from *wild-type* larvae (**e,n**) or from larvae expressing *dome^{RNAi}* under the control of the *sd-gal4* driver (**f,g,o,p**) labelled to visualize *vn* expression (*vein-lacZ*, X-Gal staining, blue) in second (**e–g**) and late third instar (**n–p**) stages. (**k–m**) Wing primordia from late third instar *wild-type* larvae (**k**) or from larvae expressing *dome^{RNAi}* under the control of the *sd-gal4* driver (**l,m**) labelled to visualize *apterous* expression (*apterous-lacZ*, X-Gal staining, blue). Scale bars, 20 μm (**c–g**) or 50 μm (**a,b,h–p**).

and very often induced ectopic wing structures emerging from the notum (Fig. 3e,f, white arrows). Ectopic wing structures were also observed in discs and adult flies upon overexpression of Hop in the P compartment (Supplementary Fig. 1). These results are reminiscent of the effects caused by local reduction of Vn/EGFR activity in the notum, as expression of a chimeric protein between Vn and the secreted EGFR antagonist Argos (Vn::Aos) also induced ectopic wing structures in the same location (Fig. 3g, see also ref. 7). All these results indicate that Upd has an early role in restricting Vn/EGFR targets to the most proximal region of the wing primordium and, by doing so, JAK/STAT ensures Wg-mediated wing fate specification (Fig. 3n).

**JAK/STAT restricts the expression of Upd**. We observed that RNAi-mediated depletion of Domeless or STAT92E in the A compartment of the wing caused in the nearby P compartment a non-autonomous increase in the levels of the 10xSTAT-GFP reporter (Fig. 3h,i, red arrow), a downregulation of *mirror* (Fig. 3j, red arrow) and the induction of wing structures, monitored by the ectopic expression of Nub (Fig. 3k, see also ref. 18). Depletion of *wg* did not rescue the non-autonomous induction of wing structures caused by *stat92E^{RNAi}* (Supplementary Fig. 1). Interestingly, expression of a truncated Domeless receptor (Dome^{DN}),

lacking the intracellular domain but able to trap Upd, did not have any non-autonomous effect (Fig. 3l). These observations and the fact that ectopic activation of JAK/STAT generates ectopic wings suggest that Upd might be either ectopically expressed or its levels increased upon JAK/STAT depletion. Consistently, expression of Dome^{DN} fully rescued the non-autonomous induction of wing structures caused by *dome^{RNAi}* (Fig. 3m, *dome^{RNAi}* targets the region encoding for the C-terminal intracellular domain and should not affect the expression levels of the *dome^{DN}* transgene, see Methods). We then analysed the expression of *upd* by *in situ* hybridization in *wild-type* and JAK/STAT-depleted discs. The early expression of *upd* in the distal domain of second instar wing discs (Supplementary Fig. 1) is later repressed in the wing pouch and restricted to five dots located in the wing hinge of mature discs (Fig. 3o). In JAK/STAT-depleted discs, the restriction of *upd* expression to these five dots was impaired and *upd* expression levels were increased (Fig. 3o, red arrows). Consistently, clones of cells mutant for *stat92E* failed to lose *upd* expression in the wing pouch (Supplementary Fig. 1). Taken together, these observations indicate that the negative feedback loop between JAK/STAT and its ligand contribute to restrict the expression levels and pattern of Upd to the maturing wing hinge (Fig. 3n) and that a failure to do so interferes with the wing versus body wall subdivision. In second instar discs,

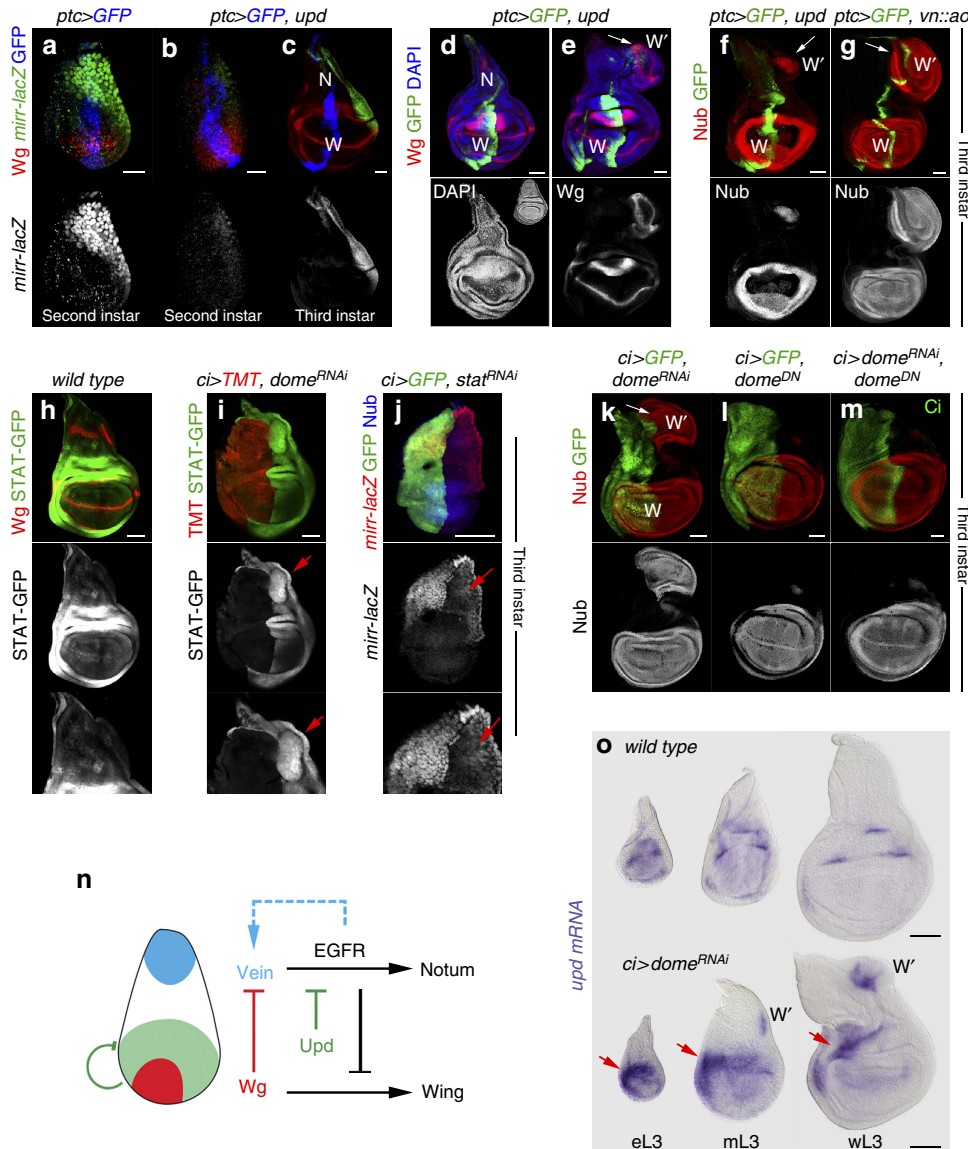

**Figure 3 | JAK/STAT deregulation leads to ectopic wing structures.** (**a–g**) Wing primordia from second (**a,b**) and late third instar (**c–g**) larvae expressing the indicated transgenes under the control of the *ptc-gal4* driver, labelled to visualize *mirror* (*mirr-lacZ*, antibody against β-gal, green or white, **a–c**), Wg (red or white, **a–e**), GFP (blue, (**a–c**), green, **d–g**), Nub (red or white, **f,g**) and DAPI (blue or white, **d,e**). In **c–g**, wing territories (W), endogenous nota (N) and ectopic wing territories (W', and white arrows) are marked. Inset in the lower panel of **d** shows a *wild-type* third instar wing disc. (**h–j**) Late (**h,i**) and early third instar (**j**) wing primordia from *wild-type* larvae (**h**) and from larvae expressing *dome^RNAi* (**i**) or *stat92E^RNAi* (**j**) under the control of the *ci-gal4* driver and labelled to visualize the activity of JAK/STAT pathway (*10xSTAT-GFP*, green or white, **h,i**), Wg (red, **h**), *Myristoylated-Tomato* (TMT, red, **i**), *mirror* (*mirr-lacZ*, antibody against β-gal, red or white, **j**), Nub (blue, **j**) and GFP (green, **j**). Red arrows point to non-autonomous ectopic activation of *10xSTAT-GFP* (**i**) and non-autonomous reduction of *mirror* expression levels (**j**). (**k–m**) Late third instar wing primordia from larvae expressing the indicated transgenes under the control of the *ci-gal4* driver labelled to visualized Nub (red or white), GFP (green, **k,l**) and Ci (green, **m**). In **k**, the endogenous (W) and ectopic (W') wing fields are marked by white arrows. (**n**) Cartoon depicting the role of Upd and the JAK/STAT signalling pathway in restricting EGFR activity to the body wall and facilitating Wg-mediated wing fate specification in early development. (**o**) Early (eL3), mid (mL3) and late (wL3) third instar wing primordia of the indicated genotypes and labelled to visualize *upd* mRNA (purple). Note that the *upd* expression pattern failed to resolve into its characteristic five-spot pattern and accumulates at higher levels in the anterior hinge (arrows). Scale bars, 20 μm (**a,b**) or 50 μm (**c–o**).

restricted Upd expression and JAK/STAT activity to the most distal part of the wing do not rely on Wg and Vn/EGFR pathways, as they were largely unaffected by *wg* loss or by ubiquitous EGFR activation (Supplementary Fig. 1). JAK/STAT depletion did not induce the ectopic expression of *upd* in the body wall of early instar discs either (Fig. 3o and Supplementary Fig. 1). Thus, the mechanism by which the early expression of *upd* is restricted to the distal wing primordium remains to be elucidated.

**JAK/STAT maintains the size of the Hh-expressing compartment.** Upd expression evolves as wing development proceeds and becomes restricted to the presumptive wing hinge—a region that connects the developing wing to the surrounding body wall[18,19,22,23] (see also Figs 3o and 8a). The 10xSTAT-GFP reporter is consequently activated in the hinge region (Figs 3h and 8b). Interestingly, mild activation of 10xSTAT-GFP was also observed in the whole wing field and this expression depended on

JAK/STAT activity, as Domeless depletion induced a cell-autonomous downregulation of the reporter (Fig. 3i and Supplementary Fig. 2). These observations prompted us to analyse whether JAK/STAT has a broader developmental role during limb development, besides its reported activity in defining and promoting wing hinge growth[19,23]. For this purpose, and in order to bypass the earlier requirement of JAK/STAT in wing fate specification, we used genetic tools with a milder effect on the pathway. Expression of Dome[DN] in the *sd-gal4* domain gave rise to a reduction in the size of the wing disc (Fig. 4a,b) with a clear impact on the size of the P compartment (Fig. 4b, white arrow). A similar phenotype was observed in *dome[RNAi]*-expressing wing discs (Supplementary Fig. 2). In some cases, the P compartment was completely lost, giving rise to a stronger decrease in the size of the wing pouch (Fig. 4c, white arrow). A similar undergrowth of the wing disc was observed in *hop[27]* mutant animals (Fig. 4d, see also ref. 21), and in this background the size of the P

compartment was also reduced (Fig. 4d, white arrow). While depletion of Dome activity in A cells (with the *ci-gal4* driver) did not have any noticeable impact on the size of the A compartment (Fig. 3i–m, Supplementary Fig. 2 and Fig. 8d), expression of Dome[DN] in P cells (with the *en-gal4* driver) caused a strong reduction in the size of the P compartment (Fig. 4e, white arrow). This reduction was also observed with RNAi forms of *dome*, *hop* and *stat92E*, as well as in JAK/STAT-depleted early third instar wing discs and mature haltere and leg primordia (Supplementary Fig. 2). To analyse whether JAK/STAT is required to maintain the size of the P compartment throughout development, we used the thermosensitive version of the Gal4 repressor, Gal80[ts], to temporally control JAK/STAT activity. The reduction in the size of the P compartment observed in early third wing primordia grown at the restrictive temperature (29 °C) was restored when larvae were shifted to the permissive temperature (18 °C) during third instar (Supplementary Fig. 2). These results indicate that

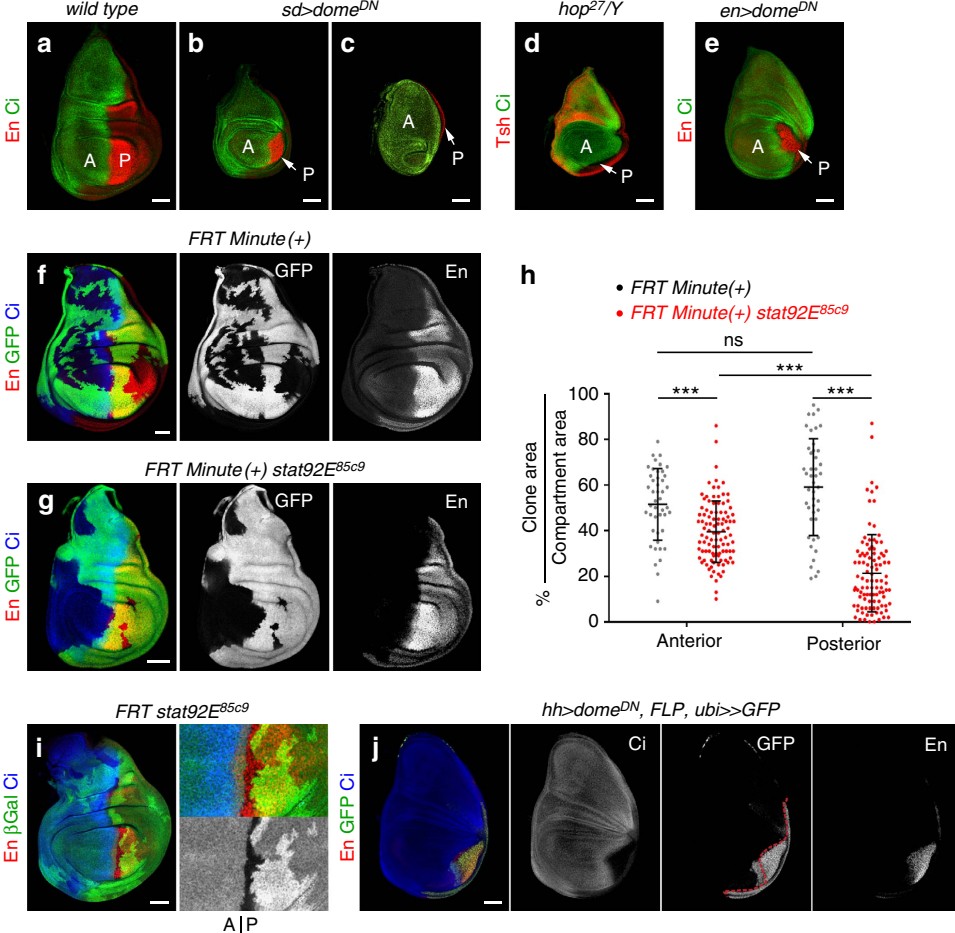

**Figure 4 | A specific requirement of JAK/STAT signalling in P cells.** (**a–e**) Late third instar wing primordia from *wild-type* (**a**) larvae or from larvae expressing the indicated transgenes under the control of the *sd-gal4* (**b,c**) or *en-gal4* (**e**) drivers, or hemizygous for the *hop[27]* allele (**d**). Wing primordia were labelled to visualize expression of Engrailed (En, red, **a–c,e**), Ci (green) to label the posterior (P) and anterior (A) compartments, respectively, and Tsh (red, **d**). Note the reduction in the size of the P compartment upon depletion of the JAK/STAT signalling pathway (white arrows). (**f,g,i**) Representative examples of mature larval wing primordia with clones of cells lacking *stat92E* activity (**g,i**) or *wild-type* for *stat92E* (**f**) labelled by absence of GFP (green or white, **f,g**) or *lacZ* expression (antibody against β-gal, green or white, **i**) and induced at first instar (see Methods). Wing primordia were stained for Engrailed (En, red or white) and Ci (blue). In **f,g**, clones were generated by the *Minute* technique. In **i**, mutant clones and their corresponding twins (labelled with two copies of *lacZ*) can be monitored. (**h**) Scatter plot showing the percentage of the A or P compartments covered by *stat92E* mutant (red) or *wild-type* (black) clones. *Wild-type* clones, anterior = 52 ± 16%, posterior = 59 ± 21% (n = 45 discs); *stat92E* clones, anterior = 40 ± 13%, posterior = 21 ± 17% (n = 102 discs). Error bars (vertical lines) represent s.d. and long horizontal lines label the average values. *P* values: Student's *t*-test, ns, not significant; ***P < 0.001. Each dot represents a wing disc. (**j**) G-TRACE mediated cell lineage analysis to irreversibly label all cells born in the P compartment upon JAK/STAT depletion in P cells. Wing disc was stained for Engrailed (En, red or white), Ci (blue and white) and GFP (green or white). Dashed red line marks the AP boundary. Scale bars, 50 μm.

JAK/STAT is required to maintain the size of the P compartment throughout development.

To further characterize the requirement of JAK/STAT in the maintenance of P compartment size, we analysed the size and distribution of clones of cells mutant for a $stat92E$ null allele ($stat92E^{85c9}$). Since cells mutant for JAK/STAT are eliminated through cell competition[22]—a process by which slow-growing cells are removed through apoptosis by fast-growing cells[32]—we gave the $stat92E^{85c9}$ mutant cells a relative growth advantage using the $Minute$ technique to impair growth of the surrounding non-mutant cells. In a $Minute/+$ background, $wild-type$ clones were similarly recovered in A and P compartments (Fig. 4f), and the average percentage of each compartment covered by these clones was largely similar (Fig. 4h). In contrast, a low number of $stat92E^{85c9}$ mutant cells were recovered in the P compartment (Fig. 4g). The average percentage of each compartment covered by $stat92E^{85c9}$ mutant clones was smaller when compared with $wild-type$ clones, and this difference was highest in the case of the P compartment (Fig. 4h). The asymmetric recovery of $stat92E^{85c9}$ mutant cells was not caused by changes in AP identity, as P mutant cells continued to express the P-selector gene $engrailed$ (Fig. 4g,i). Neither was it a consequence of cells crossing from the P to the A compartment, since clones mutant for $stat92E^{85c9}$ born in the P territory respected the AP compartment boundary (Fig. 4i). We also carried out a lineage-tracing experiment to irreversibly label all cells born in the P compartment upon Dome$^{DN}$ expression. Although a small number of cells with a P compartment origin crossed to the A compartment under these circumstances (Fig. 4j), this violation does not explain the observed reduction in the size of the P compartment. This cellular behaviour resembles the boundary transgressions observed in regenerating wing discs upon transient pro-apoptotic gene induction[33], thereby suggesting that reduced survival cues might explain the size reduction of the P compartment (see below). This reduction was not due to the transformation of the posterior wing to the posterior notum, as the characteristic expression pattern of wing and body wall markers (Nub and Tsh, respectively) was unaffected in $hop^{27}$ hemizygous mutant animals (Fig. 4d) and in Dome$^{DN}$-expressing wing discs (Supplementary Fig. 2). Altogether, these results reveal a cell-autonomous and compartment-specific requirement of JAK/STAT in promoting the growth, proliferation and/or cell survival of P cells during development.

**JAK/STAT promotes the cycling and survival of P cells**. On the basis of the cell-autonomous requirement of JAK/STAT in P cells, we monitored the activity of the major growth-promoting pathways, the expression of cell cycle markers and the activity of the apoptotic pathway in P wing cells upon JAK/STAT depletion. Growth of the developing wing relies, among others, on the activity of Dpp expressed along the AP compartment boundary, on the activity of the Hippo/Yorkie signalling pathway and on the proto-oncogene dMyc. The levels of dMyc protein, and the activity of Dpp and Hippo/Yorkie signalling pathways, monitored by the expression of its targets Spalt[34] and Expanded[35], respectively, were unaffected in P cells expressing Dome$^{DN}$ (Supplementary Fig. 3). These results are consistent with the fact that JAK/STAT mutant cells are eliminated through cell competition in a Yorkie- and dMyc-independent manner[22] and that $stat92E^{85c9}$ mutant clones were hardly recovered in the P compartment in spite of being conferred a growth advantage with the $Minute$ technique (Fig. 4g,h). We thus monitored the activity of the apoptotic pathway. A TdT-mediated dUTP nick end labeling (TUNEL) assay revealed an increase in the number of apoptotic cells in the P compartment of $hop^{27}$ wing discs

during development when compared with $wild-type$ controls (Fig. 5a–d). The $Drosophila$ inhibitor of apoptosis dIAP1 protects cells from apoptosis by inhibiting active caspases, and STAT92E, when activated, regulates $dIAP1$ expression in imaginal discs[36]. Consistently, overexpression of Hop led to a cell-autonomous increase in the expression of a $dIAP1$ enhancer trap (Fig. 5e,g). However, the role of JAK/STAT in maintaining physiological levels of dIAP1 is specific to the P compartment, as JAK/STAT depletion gave rise to a clear reduction in the levels of dIAP1 in P (compare Fig. 5h,e and Fig. 5i,f) but not in A cells (Supplementary Fig. 4). Most interestingly, overexpression of dIAP1 or an RNAi against the initiator Caspase Dronc rescued the Dome$^{DN}$-mediated size reduction of the P compartment (Fig. 5j–l and Supplementary Fig. 4) and the amount of cell death (Supplementary Fig. 4). A similar rescue of P compartment size was observed upon expression of the effector Caspase inhibitor P35 (Supplementary Fig. 4). We next analysed the expression of G1/S and G2/M rate-limiting Cyclins in P cells expressing Dome$^{DN}$. Although CycE levels were not affected (Supplementary Fig. 3), CycA and B were visibly reduced in P cells depleted of Dome activity (Fig. 5m and Supplementary Fig. 3, see also ref. 21). Expression of Dome$^{DN}$ in A cells did not cause any overt downregulation of these two G2 cyclins (Supplementary Fig. 3). Interestingly, overexpression of CycA was able to largely rescue the reduction in the P compartment size caused by Dome$^{DN}$ (Fig. 5n). This observation therefore indicates that the downregulation of this G2 cyclin is partially responsible for the reduction in the size of the P compartment caused by loss of JAK/STAT. Surprisingly, CycB overexpression did not rescue the size reduction of this compartment (Supplementary Fig. 3). Altogether, these results indicate that JAK/STAT maintains the size of the P compartment by regulating CycA and dIAP1 levels.

**JAK/STAT counteracts the activity of Engrailed**. Stable subdivision of the wing primordium into A and P compartments is a consequence of asymmetric signalling by Hh from P to A cells. The activity of En in P cells helps to generate this asymmetry by inducing the expression of Hh in the P compartment and at the same time repressing the essential downstream component of the Hh pathway Cubitus interruptus (Ci, ref. 9). Thus, only A cells that receive the Hh signal across the compartment boundary will respond by stabilizing Ci. We thus analysed whether the specific requirement of the P compartment for JAK/STAT to drive cell cycling and survival can be explained by the absence of Hh signalling or the presence of En in these cells. In the first case, the combined activities of two signalling molecules (Upd and Hh) might promote survival and proliferation of A cells, whereas the sole action of Upd through JAK/STAT might exert a similar role in P cells. However, depletion of Hh signalling together with JAK/STAT in A cells did not cause any obvious phenotype in terms of compartment size (Supplementary Fig. 5). We next addressed the alternative hypothesis and tested whether JAK/STAT counteracts the negative effects of the En transcriptional repressor in cell cycling and survival. A reduction in the levels of En in P cells (Supplementary Fig. 5), either by expression of two independent $en^{RNAi}$ transgenes or by halving the doses of $en$ and $invected$ genes (in $Df(2)en^E/+$ individuals), substantially rescued the reduction in the size of the P compartment caused by Dome$^{DN}$ (Fig. 6b–d and Supplementary Fig. 5). Most interestingly, the reduction in CycA levels and the amount of cell death observed in JAK/STAT-depleted P compartments were both largely rescued upon expression of $en^{RNAi}$ (Fig. 6a–c and Supplementary Fig. 5). The rescue in tissue size, apoptosis and CycA protein levels caused by expression of $en^{RNAi}$ was not an indirect consequence of a reduction in the expression of the $en-gal4$ driver. If anything,

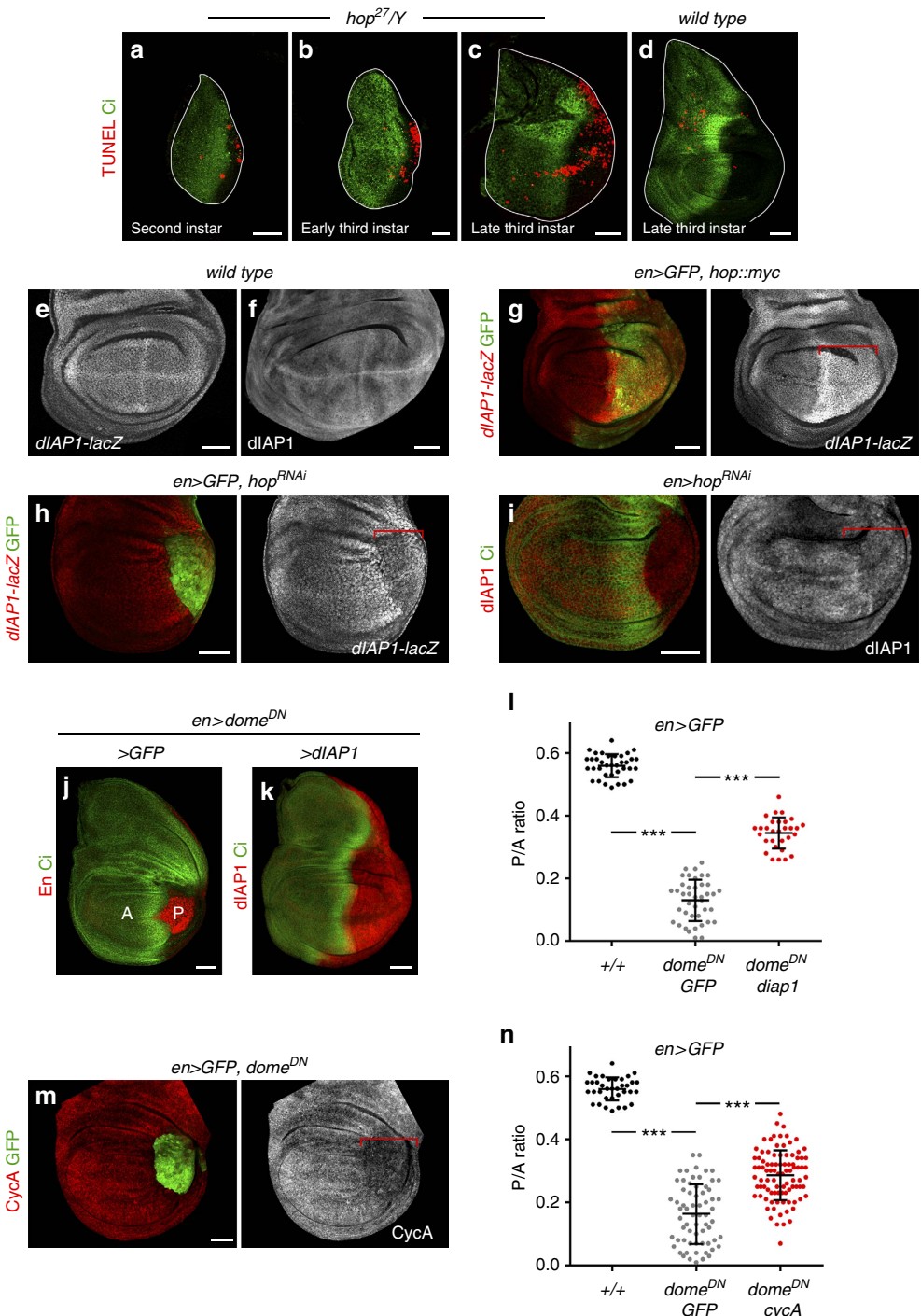

**Figure 5 | JAK/STAT promotes survival and cycling of P cells. (a–d)** Wing primordia from larvae of the indicated genotypes dissected at second (**a**), early third (**b**) or late third (**c,d**) instars, and labelled to visualize apoptotic cells by TUNEL staining (red) and Ci (green) to label the A compartment. The contour of the wing discs is marked by a white line. (**e–i**) Late third instar wing primordia from larvae of the indicated genotypes labelled to visualize expression of the *dIAP1-lacZ* enhancer trap (red or white, **e,g,h**) and dIAP1 protein levels (red or white, **f,i**). Wing discs were labelled either with Ci (green, **i**) or GFP (green, **g,h**), which mark the A and P compartments, respectively. Red brackets mark the domain of transgene expression. (**j,k**) Late third instar wing primordia from larvae of the indicated genotypes labelled to visualize Ci (green), En (red, **j**) or dIAP1 (red, **k**) protein expression. (**m**) Late third instar wing primordium of the indicated genotype labelled to visualize Cyclin A (CycA, red or white) and GFP (green). Red bracket marks the domain of transgene expression. Scale bars, 20 μm (**a,b**) or 50 μm (**c-k,m**). (**l,n**) Scatter plots showing the P/A ratio of wing primordia expressing the indicated transgenes under the control of the *en-gal4* driver. In **l**, P/A ratios from left to right: 0.56 ± 0.04 (*n* = 36); 0.13 ± 0.07 (*n* = 41); 0.34 ± 0.05 (*n* = 30). In **n**, P/A ratios from left to right, 0.56 ± 0.04 (*n* = 36); 0.16 ± 0.09 (*n* = 66); 0.29 ± 0.08 (*n* = 92). Error bars (vertical lines) represent s.d. and long horizontal lines mark the average values. *P* values: Student's *t*-test, \*\*\**P* < 0.001.

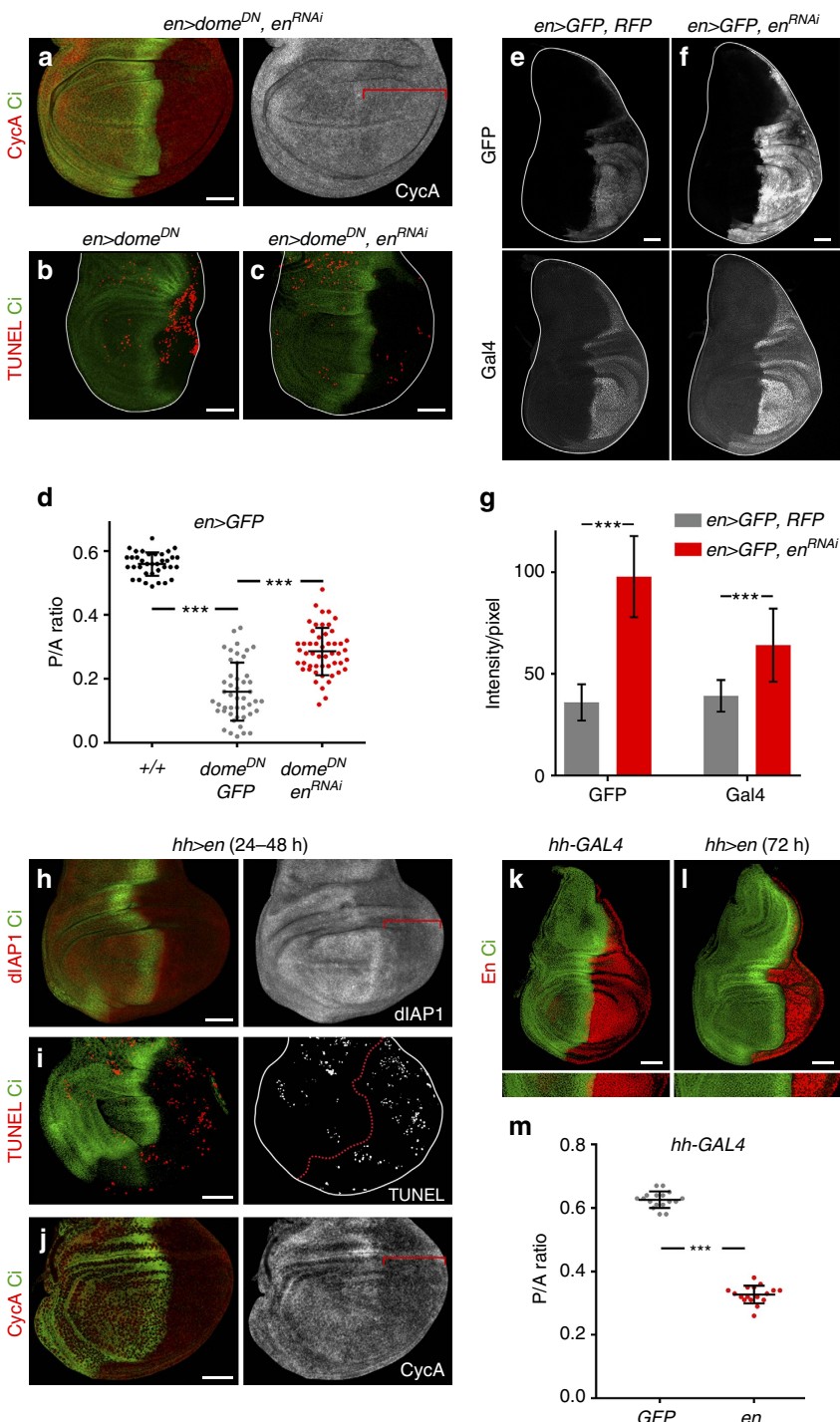

**Figure 6 | JAK/STAT counteracts En activity on survival and cycling of P cells.** (**a**–**c**) Late third instar wing primordia from larvae of the indicated genotypes labelled to visualize Cyclin A (CycA, red or white, **a**), apoptotic cells by TUNEL staining (red, **b**,**c**) and Ci (green). (**e**,**f**) Late third instar wing primordia from larvae of the indicated genotypes labelled to visualize GFP or Gal4 (white). (**g**) Bar graph showing the average signal intensity/pixel of GFP and Gal4 protein levels of the indicated genotypes. For $en > GFP$, $RFP$, GFP intensity $= 35.93 \pm 8.93$; Gal4 intensity $= 39.14 \pm 7.79$ ($n = 18$). For $en > GFP$, $en^{RNAi-VDRC}$, GFP intensity $= 97.86 \pm 19.88$; Gal4 intensity $= 64.13 \pm 18.27$ ($n = 23$). Error bars represent s.d. P values: Student's t-test, ***$P < 0.001$. (**h**–**l**) Late third instar wing primordia from individuals overexpressing En (**h**–**j**,**l**) or GFP (**k**) in the P compartment for 24–48 h (**h**–**j**), or 72 h (**k**,**l**), and labelled to visualize dIAP1 protein (red or white, **h**), apoptotic cells by TUNEL staining (red or white, **i**), CycA (red or white, **j**), En (red, **k**,**l**) and Ci (green) protein expression. Red brackets in **a**,**h**,**j** mark the domain of transgene expression; wing disc contours are marked in **b**,**c**,**e**,**f**,**i** by a white line; red dashed line in **i** marks the AP boundary. Scale bars, 50 μm. (**d**,**m**) Scatter plots showing the P/A ratio of wing primordia expressing the indicated transgenes under the control of the $en$-$gal4$ (**d**) or $hh$-$gal4$ (**m**) drivers. In **d**, P/A ratios from left to right: $0.56 \pm 0.04$ ($n = 36$); $0.16 \pm 0.09$ ($n = 47$); $0.29 \pm 0.07$ ($n = 51$). In **m**, P/A ratios from left to right: $0.63 \pm 0.03$ ($n = 17$); $0.33 \pm 0.03$ ($n = 17$). Error bars (vertical lines) represent s.d. and long horizontal lines mark average values. P values: Student's t-test, ***$P < 0.001$.

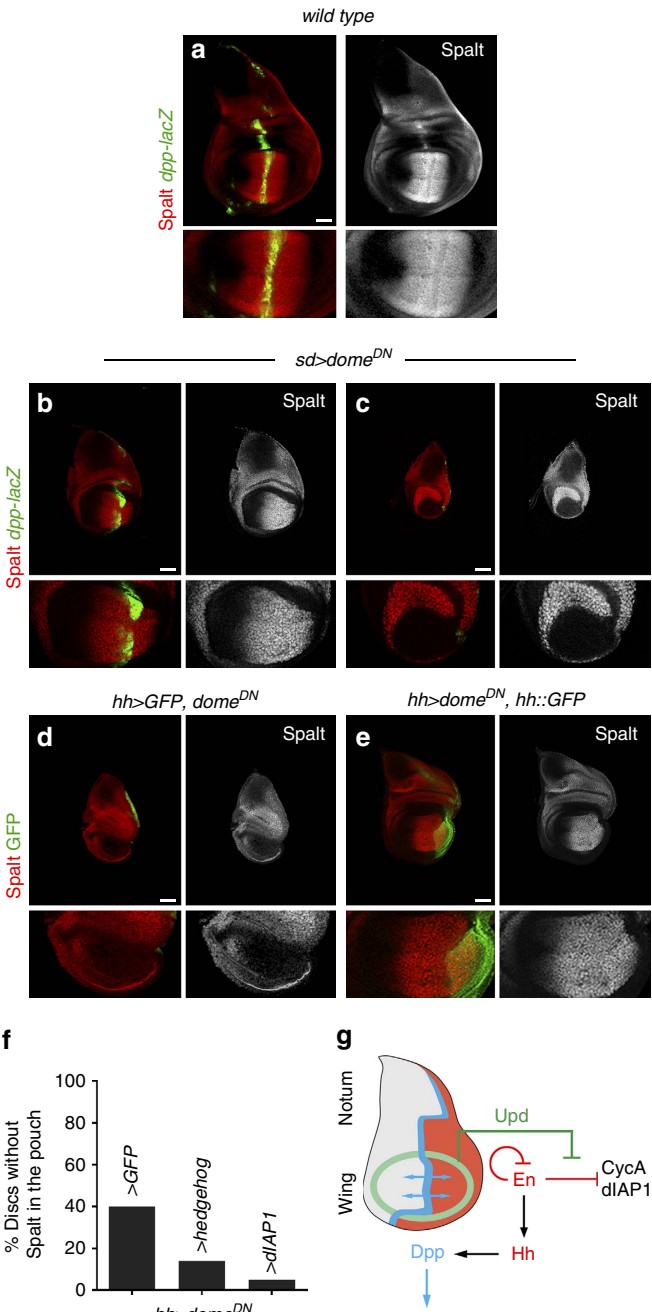

**Figure 7 | JAK/STAT promotes Dpp expression by maintaining the pool of Hh-expressing cells.** (a–e) Late third instar wing primordia from larvae of the indicated genotypes labelled to visualize expression of *dpp-lacZ* (antibody against β-gal, green, **a–c**), GFP (green, **d,e**) and Spalt (red or white). Higher magnifications of the wing pouch are shown in the lower panels. Scale bars, 50 μm. (**f**) Bar graph plotting the percentage of wing primordia in which Dpp was lost monitored by the absence of Spalt expression in the wing pouch. Wing primordia are expressing the indicated transgenes under the control of the *hh-gal4* driver. From left to right: 40% (n = 53); 14% (n = 43); 5% (n = 42). (**g**) Cartoon depicting the role of Upd and the JAK/STAT signalling pathway in Dpp-dependent wing growth by promoting the survival and cycling of Hh-producing cells.

the expression levels of this driver increased, monitored by an *UAS-GFP* transgene and antibody to the Gal4 protein (Fig. 6e–g). This observation is consistent with the reported capacity of En to negatively regulate its own expression that is used to finely modulate physiological En expression levels[37]. Moreover,

overexpression of En in its domain gave rise to a clear reduction in dIAP1 and CycA protein levels, caused apoptotic cell death and reduced the size of the P compartment (Fig. 6h–m). Collectively, these results indicate that JAK/STAT counteracts the negative impact of Engrailed on the cycling and survival of P cells.

**Stable localization of the Dpp organizer relies on JAK/STAT.** Hh from P cells induces the expression of Dpp in A cells abutting the P compartment, and Dpp organizes the growth and patterning of the developing appendage[3]. We thus analysed whether the strong reduction in the pool of Hh-expressing cells caused by Dome[DN] expression had any impact on Dpp expression and, consequently, on wing growth. As noted above, two distinct growth phenotypes could be observed in wing discs expressing Dome[DN] in the *sd-gal4* domain (Fig. 4b,c). In most cases, a mild but reproducible growth defect was accompanied by a clear size reduction of the P compartment (Fig. 4b), and in all these cases the expression of Dpp and its target gene *spalt* was maintained (Fig. 7a,b). However, a certain fraction of wing discs showed a complete loss of the P compartment, accompanied by a strong reduction in the size of the wing pouch (Fig. 4c), resembling the tissue size defects observed in *dpp* mutant wing discs[12]. Consistent with the reduction in the number of Hh-producing cells, the stripe of Dpp expression and its downstream target gene *Spalt* were lost in these cases (Fig. 7c). These two distinct phenotypes were also obtained by expressing Dome[DN] in the P compartment (Fig. 7d and Supplementary Fig. 3). Remarkably, the fraction of wing discs that lost Dpp activity, visualized by the expression of Spalt, was clearly reduced upon overexpression of Hh or dIAP1 in the P compartment (Fig. 7d–f). Thus, the pro-survival and mitogenic activity of JAK/STAT signalling in P cells contributes to the maintenance of a pool of Hh-producing cells that induce Dpp expression in nearby A cells, thus giving rise to well-sized and fully functional limb primordia (Fig. 7g).

**JAK/STAT restricts the Dpp organizer to the developing limb.** High levels of Upd expression and JAK/STAT activity in the hinge primordium (Fig. 8a,b) contribute to its growth[18,19,23]. Consistently, expression of Dome[DN] in the A compartment gave rise to a reduction in hinge size and to the close apposition of the developing appendage and the surrounding body wall or notum (Fig. 8c,d, red brackets). Dpp is expressed at the AP boundary; however, it exerts its organizing activities only in the wing appendage (Fig. 8e). We thus wondered whether the hinge region acts as a fence that contributes to isolating the organizing activity of Dpp to the developing appendage. Interestingly, in primordia in which Dome[DN] was expressed in the hinge, the AP boundary of the body wall became closer to the developing wing (Fig. 8c,f, white arrows), the Dpp target gene *Spalt* was ectopically induced in nearby wing cells (Fig. 8g, white star), the wing pouch was expanded (Fig. 8h, white star) and pattern duplications in the P compartment of the adult wings were frequently observed (Fig. 8i–k, red stars). Altogether, these results indicate that JAK/STAT signalling contributes, through its growth-promoting activity in the hinge region, to isolate the body wall and appendage sources of Dpp, thus restricting the organizing activity of Dpp to the developing appendage.

**Discussion**
Morphogens of the Wnt/Wg, Shh/Hh and BMP/Dpp families regulate tissue growth and pattern formation in vertebrate and invertebrate limbs. Here we unravel a fundamental role of the secreted Upd ligand and the JAK/STAT pathway in facilitating the activities of these three morphogens in exerting their fate- and growth-promoting activities in the *Drosophila* wing primordium.

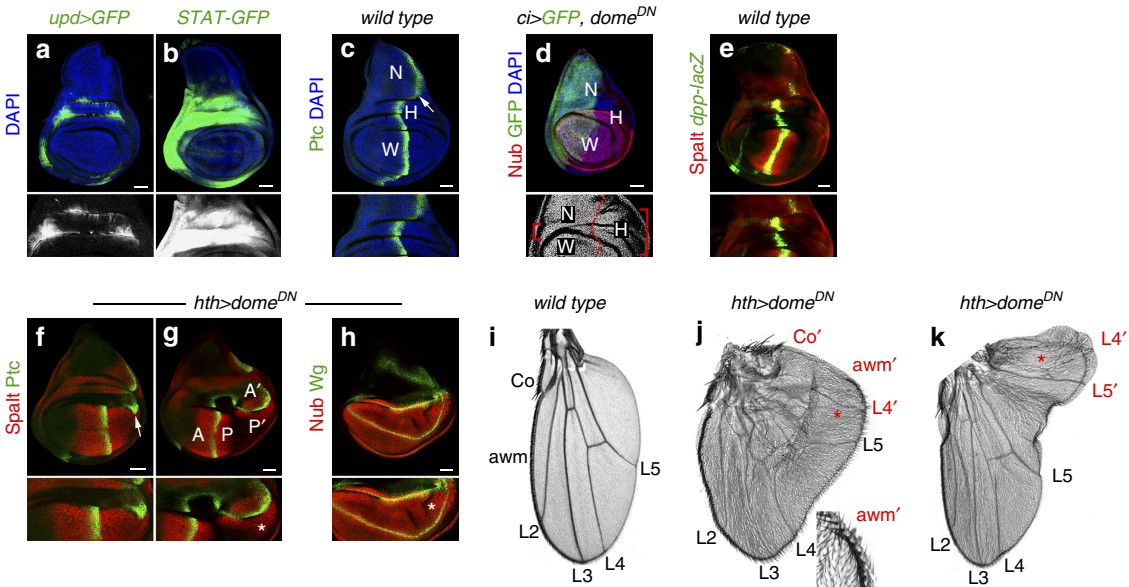

**Figure 8 | JAK/STAT restricts the Dpp organizer to the developing appendage.** (**a–c**) Late third instar wing primordia labelled to visualize expression of *upd* (*upd-gal4, UAS-myrGFP*, green or white, **a**), activity of the JAK/STAT pathway (*10xSTAT-GFP*, green or white, **b**), Ptc (green, **c**) to mark the AP compartment boundary and DAPI (blue). Magnifications of the hinge region, where JAK/STAT signalling is highly active, are shown in the lower panels. White arrow marks the AP boundary of the body wall in **c**. (**d**) Depletion of JAK/STAT in the A compartment compromises hinge growth. GFP (green) marks the A compartment and Nub (red) the wing pouch. DAPI is shown in blue. Red brackets in the lower panel show the affected hinge region in the A compartment compared with the P compartment. Dashed red line marks the AP boundary. Wing (W), notum (N) and hinge (H) territories are marked in **c,d**. (**e**) Late third instar wing primordium labelled to visualize expression of Spalt (red) and *dpp-lacZ* (antibody against β-gal, green). (**f–h**) Late third instar wing primordia expressing *dome*^DN under the control of the *hth-gal4* driver and labelled to visualize Spalt (red, **f,g**), Ptc (green, **f,g**), Nub (red, **h**) and Wg (green, **h**). White arrow in **f** marks the AP boundary of the body wall abutting the wing pouch as a consequence of the reduction in hinge size, and stars in **g,h** mark the ectopic expression of Spalt (**g**) and the resulting wing outgrowth marked by Nub (**h**). Scale bars, 50 μm. (**i–k**) Cuticle preparations of adult wings from *wild-type* individuals (**i**) or from individuals expressing *dome*^DN under the control of the *hth-gal4* driver (**j,k**). L2–L5, longitudinal veins; Co, costa; awm, anterior wing margin. The wings of *hth>dome*^DN individuals present pattern duplications of the P structures (for example, L4′, L5′), and in these duplications A structures (awm′ and Co′) are often observed. Higher magnification of the ectopic awm (awm′) is shown in the inset of **j**.

Early in wing development, two distinct mechanisms ensure the spatial segregation of two alternative cell fates. First, the proximal–distal subdivision of the wing primordium into the wing and the body wall relies on the antagonistic activities of the Wg and Vn signalling molecules. While Wg inhibits the expression of Vn and induces the expression of the wing-determining genes, Vn, through the EGFR pathway, inhibits the cellular response to Wg and instructs cells to acquire body wall fate[7]. Second, growth promoted by Notch pulls the sources of expression of these two morphogens apart, alleviates the repression of wing fate by Vn/EGFR, and contributes to Wg-mediated appendage specification[8]. Expression of Vn is reinforced by a positive amplification feedback loop through the activation of the EGFR pathway (Fig. 3n, ref. 7,31). This existing loop predicts that, in the absence of additional repressors, the distal expansion of Vn/EGFR and its targets would potentially impair wing development[7]. Our results indicate that Upd and JAK/STAT restrict the expression of EGFR target genes and Vn to the most proximal part of the wing primordium, thereby interfering with the loop and allowing Wg to correctly trigger wing development. We also present evidence that JAK/STAT restricts the expression pattern and levels of its own ligand Upd and that ectopic expression of Upd is able to bypass EGFR-mediated repression and trigger wing development *de novo*. This negative feedback loop between JAK/STAT and its ligand is of biological relevance, since it prevents high levels of JAK/STAT signalling in proximal territories that would otherwise impair the development of the notum or cause the induction of supernumerary wings, as shown by the effects of ectopic

activation of the JAK/STAT pathway in the proximal territories. Thus, while Wg plays an instructive role in wing fate specification, the Notch and JAK/STAT pathways play a permissive role in this process by restricting the activity range of the antagonizing signalling molecule Vn to the body wall region (Fig. 3n).

Later in development, once the wing field is specified, restricted expression of Dpp at the AP compartment boundary organizes the growth and patterning of the whole developing appendage (Fig. 7g; ref. 3). Dpp expression is induced in A cells by the activity of Hh coming from P cells, which express the En transcriptional repressor[12,38]. Here we show that JAK/STAT controls overall organ size by maintaining the pool of Hh-producing cells to ensure the stable and localized expression of the Dpp organizer. JAK/STAT does so by promoting the cycling and survival of P cells through the regulation of *dIAP1* and CycA, counteracting the negative effects of En on these two genes. Since the initial demonstration of the role of the AP compartment boundary in organizing, through Hh and Dpp, tissue growth and patterning, it was noted that high levels of En interfered with wing development by inducing the loss of the P compartment[38,39]. The capacity of En to negatively regulate its own expression was subsequently shown to be mediated by the Polycomb-group genes and proposed to be used to finely modulate physiological En expression levels[37]. Consistent with this proposal, we observed an increase in the expression levels of the *en-gal4* driver, which is inserted in the *en* locus and behaves as a transcriptional reporter, in *en*^RNAi-expressing wing discs. The negative effects of En on cell cycling and survival reported in our

work might also contribute to the observed loss of the P compartment caused by high levels of En. As is it often the case in development, a discrete number of genes is recurrently used to specify cell fate and regulate gene expression in a context-dependent manner. We propose that the capacity of En to block cell cycle and promote cell death might be required in another developmental context and that this capacity is specifically suppressed in the developing *Drosophila* limbs by JAK/STAT, and is modulated by the negative autoregulation of En, thus allowing En-dependent induction of Hh expression and promoting Dpp-mediated appendage growth (Fig. 7g). It is interesting to note in this context that En-expressing territories in the embryonic ectoderm are highly enriched in apoptotic cells[40]. Whether this apoptosis plays a biological role and relies on En activity requires further study.

Specific cell cycle checkpoints appear to be recurrently regulated by morphogens and signalling pathways, and this regulation has been unveiled to play a major role in development. Whereas Notch-mediated regulation of CycE in the *Drosophila* eye and wing primordia is critical to coordinate tissue growth and fate specification by pulling the sources of two antagonistic morphogens apart[8,41], our results indicate that JAK/STAT-mediated regulation of CycA is critical to maintain the pool of Hh-producing cells in the developing wing and to induce stable Dpp expression. The development of the wing hinge region, which connects the developing appendage to the surrounding body wall and depends on JAK/STAT activity[18,19,23], has been previously shown to restrict the Wg organizer and thus delimit the size and position of the developing appendage[42]. Our results support the notion that JAK/STAT and the hinge region are also essential to restrict the organizing activity of the Dpp morphogen to the developing appendage. Taken together, our results reveal a fundamental role of JAK/STAT in promoting appendage specification and growth through the regulation of morphogen production and activity, and a role of pro-survival cues and mitotic cyclins in regulating the pool of morphogen-producing cells in a developing organ.

The striking parallelisms in the molecules and mechanisms underlying limb development in vertebrates and invertebrates have contributed to the proposal that an ancient patterning system is being recurrently used to generate body wall out-growths[43,44]. Whether the conserved JAK/STAT pathway plays a developmental role also in the specification or growth of vertebrate limbs by regulating morphogen production or activity is a tempting question that remains to be elucidated.

## Methods

**Drosophila Strains.** The following stocks are described in Flybase: apterous-lacZ; vein-lacZ[8]; UAS-dome$^{\Delta CYT}$ (UAS-dome$^{DN}$ in the text[45]); UAS-vn:aos[7]; upd1-gal4 (ref. 24); UAS-dIAP1 (ref. 36); UAS-upd[19]; FRT82B stat92E$^{85c9}$ (ref. 22); FRT82B stat92E$^{85c9}$ UbiRFP[46]; UAS-hop::myc[47]; UAS-hh::GFP[48]; iro$^{EGP7}$ (ref. 49); UAS-en; UAS-CycB::HA[50]; UAS-EGFR$^{\lambda TOP4.2}$ (ref. 51); sd-gal4; ci-gal4; hth-gal4; ptc-gal4; hh-gal4; vn-gal4; dpp-lacZ; en.lacZ. The following strains were provided by the Bloomington Drosophila Stock Center (BDSC) or the Vienna Drosophila RNAi Center (VDRC): UAS-dome$^{RNAi}$ (VDRC 106071 and BDSC 34618); UAS-hop$^{RNAi}$ (BDSC 32966); UAS-stat92E$^{RNAi}$ (BDSC 33637 and VDRC 106980); UAS-dronc$^{RNAi}$ (VDRC 23033); UAS-en$^{RNAi}$ (VDRC 105678 and BDSC 26752); hop[27] (BDSC 8493); 10xSTAT-GFP (BDSC 26197 and 26198); mirror-lacZ (BDSC 10880); UAS-CycA (BSDC 6633); diap1-lacZ (BDSC 12093); wg$^{CX4}$ (BDSC 2980); wg$^{CX3}$ (BDSC 2977); EGFR$^{F2}$ (BDSC 2768); UAS-sgg (BDSC 5255); en-gal4 (BDSC 30564); UAS-myristoylated-Tomato (UAS-myrT, BDSC 32222); UAS-GFP (BDSC 4775, 6658 and 6874); UAS-myristoylated-GFP (UAS-myrGFP, BDSC 32196); UAS-RFP (BDSC 30556); UAS-dcr2 (BDSC 24644 and 24651); ubi-FRT-stop-FRT-GFP (BDSC 32250); UAS-FLP (BDSC 4539); UAS-wg$^{RNAi}$ (BDSC 33902); UAS-p35 (BDSC 5072); UAS-smo5A (UAS-smo$^{DN}$ in the text, BDSC 23943); Df(2)en$^E$ (BDSC 2216).

Following the protocol described in ref. 52, RNAi strains from the VDRC KK stock collection were routinely tested for the existence of unwanted second site insertions by a diagnostic PCR and cleaned by a genetic recombination scheme. According to the VDRC, the *UAS-dome$^{RNAi}$* line used in Fig. 3m (VDRC 106071)

does not target the mRNA encoding the truncated form of *dome$^{\Delta CYT}$*. Larvae were grown in standard fly food at 29 °C to enhance RNAi-mediated gene depletion. In the case of strong gal4 drivers (for example, ci-gal4, ptc-gal4, hth-gal4, sd-gal4) larvae were generally grown at 25 °C to decrease larval and pupal lethality. See Supplementary Table 1 for fly genotypes.

**Immunohistochemistry.** Mouse anti-Wg (1:10-50; 4D4, DSHB); goat anti-Hth (1:50; sc-26187, Santa Cruz Biotechnology); mouse anti-Nub (1:10; gift from S. Cohen); rabbit anti-Nub (1:600; gift from X. Yang); mouse anti-βgal (1:50; 40-1a, DSHB); mouse anti-En (1:5; 4D9, DSHB); rat anti-Ci (1:10; 2A1, DSHB); mouse anti-Ptc (1:50; Apa1, DSHB); mouse anti-CycA (1:50; A12, DSHB); mouse anti-CycB (1:50; F2F4, DSHB); rabbit anti-CycE (1:100; sc-481, Santa Cruz Biotechnology); mouse anti-Diap1 (1:200; gift from B. Hay); rabbit anti-Tsh (1:600, gift from S. Cohen); rabbit anti-Sal (1:500, gift from R. Barrio[34]), guinea pig anti-dMyc (1:1,000; gift from G. Morata[16]); rabbit anti-Gal4 (1:100; sc-577, Santa Cruz Biotechnology); sheep anti-DIG-AP (1:2,000; 11093274910, Roche Diagnostics). Secondary antibodies Cy2, Cy3, Cy5 and Alexa 647 (1:400) were obtained from Jackson ImmunoResearch. TUNEL staining was adapted from ref. 53 with the *In Situ* Cell Death Detection Kit, TMR Red (Roche Diagnostics). *In situ* hybridization with an *upd* RNA probe (gift from F. Serras) was performed as in ref. 54.

**Quantification of tissue size.** *P/A ratio measurements of wing discs*: in the case of wing discs in which JAK/STAT was depleted in the P compartment, flies were allowed to lay eggs at 25 °C overnight, resulting larvae were transferred to 29 °C and wing discs were dissected in late third instar stages. In the case of wing discs overexpressing Engrailed in the P compartment, flies were allowed to lay eggs at 18 °C overnight, larvae were shifted to 29 °C in early second instar (4 days after egg laying, AEL) and wing discs were dissected 72 h later in late third instar stages. The size of the A compartment and of the whole wing primordium were measured using the Fiji Software[55]. The size of the P compartment was obtained by subtracting the A compartment size from the size of the whole wing disc. Number of wing discs analysed for each experiment are indicated in the corresponding figure legends. Since the P/A ratio of late third instar wing discs is largely constant, the same data set for the *wild-type* controls was used in the experiments with *en-gal4* (en-gal4, UAS-GFP/ + ; n = 36) or with *hh-gal4* (UAS-GFP/ + ; hh-gal4, tub-gal80$^{TS}$/ + ; n = 17). For each independent experiment, a control using the same number of UAS-transgenes was raised in parallel. *Clonal area measurements*: the Fiji Software[55] was used to measure the size of the A compartment and of the whole wing primordium. The size of the P compartment was obtained by subtracting the A compartment size from the size of the whole-wing disc. The clonal area that covers each compartment was obtained by measuring the area devoid of GFP expression in each domain with a macro for the Fiji Software provided by the Advanced Digital Microscopy Facility at the IRB Barcelona. The clone area/compartment area ratios were calculated. The corresponding mean and s.d.'s were calculated, and a two-tailed unpaired *t*-test assuming equal variances was carried out in Microsoft Excel. *$P < 0.05$; **$P < 0.01$; ***$P < 0.001$. Graphical representations of data were made using GraphPad Prism version 6.07.

**Quantification of cell death.** Images from basal planes were considered for the determination of the number of cells labelled by TUNEL in the P compartment, and absolute numbers of apoptotic cells were quantified with the Fiji Software[55]. All genotypes were analysed in parallel. The corresponding mean and s.d. were calculated, and a two-tailed unpaired *t*-test assuming equal variances was carried out in Microsoft Excel. *$P < 0.05$; **$P < 0.01$; ***$P < 0.001$. Graphical representations of data were made using GraphPad Prism version 6.07.

**Quantification of signal intensity.** Control ($n = 18$) and experimental ($n = 23$) wing discs were fixed and stained together to avoid variability between discs. Samples were imaged under identical settings using a Leica SP5 confocal microscope. Confocal conditions were adjusted to minimize saturated pixels with maximal intensity. To quantify GFP and Gal4 expression levels in the P compartment, average signal intensity per pixel was obtained from raw images using the histogram function of the Fiji Software. The corresponding mean and s.d. were calculated, and a two-tailed unpaired *t*-test assuming equal variances was carried out in Microsoft Excel. ***$P < 0.001$. Graphical representations of data were made using GraphPad Prism version 6.07.

**Mosaic analysis and lineage tracing.** Loss-of-function clones for the stat92E$^{85c9}$ allele were generated in the following genotypes: hs-FLP; FRT82B stat92E$^{85c9}$/ FRT82B M(3)95A2 UbiGFP (Minute+ clones), hs-FLP; FRT82B stat92E$^{85c9}$/ FRT82B arm-lacZ (twin/clone analysis) and upd-gal4, UAS-myrGFP/hs-FLP; FRT82B stat92E$^{85c9}$ UbiRFP/FRT82B. Flies were allowed to lay eggs for 4 h at 25 °C in 55 mm Petri dishes with standard food. Hatched larvae were synchronized at early first instar and allowed to grow at 25 °C in standard fly food. Sixteen hours later (40 h after egg laying, AEL), larvae were heat-shocked at 38 °C for 1 h, and wing discs were dissected ~100 h after clone induction for the Minute clones, ~85 h after clone induction for the twin/clone analysis and 24–48 h after clone

induction to monitor *upd* expression in *stat92E* mutant cells. The following genotype was used to lineage-trace the P compartment upon *dome^DN* expression: *UAS-FLP/ubi-FRT-stop-FRT-GFP; UAS-dome^DN/hh-GAL4.*

**Temporal and regional control of targeted gene expression.** Transgene expression was temporally controlled in the following experiments using the tub-GAL80^TS transgene: (1) *Ectopic expression of Upd.* To monitor *mirror-lacZ* expression, flies were allowed to lay eggs at 25 °C overnight and *ptc-gal4, UAS-GFP/UAS-upd; mirror-lacZ/tub-gal80^TS* larvae were raised at 29 °C until dissection in second or third instar larval stages. To visualize ectopic wings emerging from the notum, larvae were maintained at 18 °C, shifted to 29 °C in late second instar (5 days after egg laying, AEL) and dissected at late third instar stages. (2) *Overexpression of Shaggy.* Flies were allowed to lay eggs at 18 °C overnight, *ptc-gal4, UAS-GFP/ UAS-sgg; tub-gal80^TS/ +* larvae were shifted to 29 °C in early second instar stage (4 days after egg laying, AEL) and wing discs were dissected at late third instar stages. (3) *Temporal depletion of JAK/STAT in the P compartment.* Flies were allowed to lay eggs at 25 °C overnight, control (*UAS-GFP/ +; hh-gal4, tub-gal80^TS/ +*) and experimental (*hh-gal4, tub-gal80^TS/UAS-dome^DN*) larvae were transferred to 29 °C and wing discs were dissected at early and late third instar stages (3 and 5 days AEL, respectively). To address the capacity of the wing disc to recover P compartment size, experimental (*hh-gal4, tub-gal80^TS/UAS-dome^DN*) larvae were grown at 29 °C, shifted to 18 °C at early third instar (3 days AEL) and kept at this temperature for 4 days until wing disc dissection (at late third instar). (4) *Overexpression of Engrailed.* Flies were allowed to lay eggs at 18 °C overnight, and *hh-gal4, tub-gal80^TS/UAS-en* larvae were shifted to 29 °C in early second (4 days AEL) or mid third instar (7 days AEL) and wing discs were dissected 72 or 24–48 h later, respectively.

**Data availability.** The authors declare that all data supporting the findings of this study are available within the article and its Supplementary Information files or from the corresponding author upon reasonable request.

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

## Acknowledgements

We thank R. Barrio, S. Campuzano, J. Castelli-Gair, C. Estella, I. Guerrero, G. Halder, B. Hay, R. Holmgren, G. Morata, A. Simcox, H. Steller, F. Serras, S. Cohen, X. Yang, G. Jiménez, E. Martin-Blanco, the Bloomington *Drosophila* Stock Center (USA) and the Vienna *Drosophila* RNAi Center (Austria) for flies, the Developmental Studies Hybridoma Bank (USA) for antibodies, the Advanced Digital Microscopy Facility of the IRB for technical assistance, J.M. Murillo for the initial observations on the role of JAK/STAT in early wing development, T. Yates for text editing and Lara Barrio and Najate Benhra for comments on the manuscript. C.R.-A. was funded by a PhD fellowship from MINECO (Government of Spain), and A.F. by a PhD fellowship from the Fundação para a Ciência e a Tecnologia (SFRH/BD/68745/2010 grant). M.M. is an ICREA Research Professor. This work was funded by the *SIGNAGROWTH-BFU2013-44485* grant from MINECO (Government of Spain). IRB Barcelona is the recipient of a Severo Ochoa Award of Excellence from MINECO (Government of Spain).

## Author contributions

C.R.-A., A.F. and M.M. conceived and designed the experiments and analysed the data. C.R.-A. and A.F. performed the experiments. M.M. wrote the paper.

## Additional information

**Competing financial interests:** The authors declare no competing financial interests.

