## [Peer Review File · Nature Communications]

Reviewer #1 (Remarks to the Author)

The paper by Recanses-Alvarez et al. examines in detail the roles of the JAK/STAT pathway in wing development and identifies three distinct phases. During the 2nd instar, expression of Upd in the proximal side of the disc is required to attenuate EGFR activity, but does not play a role in defining the opposite expression domains of Wg and Vn. During the 3rd instar the JAK/STAT pathway attenuates the negative effects of En on cell division. Thus, JAK/STAT is specifically required in the posterior compartment, for its proper growth. In extreme cases when the P compartment is extremely small, there are secondary effects on growth due to low levels of Hh and low levels of Dpp. Finally, division of cells in the wing hinge, promoted by the JAK/STAT pathway, insulate the hinge from the effects of Dpp in the wing pouch, and allows its proper development.

The experiments are carefully executed, and the conclusions are well justified by experimental data. The precise definition of roles for JAK/STAT during wing development is important, and some attempts to get at the actual mechanism have been made, when relevant target genes that are affected by the JAK/STAT pathway were identified. This paper contributes important aspects to our global understanding of the stepwise processes leading to wing growth and development. Although not particularly inspiring, this paper should be published and will be of interest to a broad audience.

Reviewer #2 (Remarks to the Author)

Recasaens -Alvarez et al present a paper where they discuss the involvement of JAK/STAT signaling in the proliferation and patterning of the wing imaginal disc. The effect of stat92E on posterior compartment cells is very intriguing and is by far the strongest part of the manuscript.

STAT has been previously shown to give competitive growth advantage to cells. The work presented here shows that besides its requirement for cell competition, STAT is required in the posterior compartment of the wing to maintain cell proliferation and viability.

The manuscript is interesting but has a major fault likely to irritate many JAK/STAT signaling researchers: it does not credit sufficiently previously published work.

The first 24 lines of the introduction are too general and uninformative. They deal on the importance of morphogens without specifying which ones. It is only when they start talking about hh when they mention relevant data. In comparison, the introduction on Jak/Stat, the main topic of the manuscript, is reduced to three lines. Very relevant papers dealing on topics at the heart of this work should be cited, these include Hatini et al 2013 Dev Biol. 378: 38-50 and Rodrigues et al. 2012 Development 139:4051-4061. These are relevant papers that, as the manuscript is written, seem hidden away.

Major points:

1-Not enough credit is given to prior JAK/STAT work. Primary papers that showed very relevant results to the topic of the manuscript are ignored. For example:

1.1- Mukherjee et al Oncogene (2005) 24, 2503-2511, mentions the regulation of CyclinB by the JAK/STAT pathway and describes the small disc phenotype of hop mutant discs. Both topics are introduced in the paper without mentioning the prior work, giving the wrong impression that these observations are new.

1.2- Similar experiments to those described at the start of the Results section in Fig1c-d have been published by Hatini et al 2013 (Dev Biol) and by Rodrigues et al Development 2012 and here are reported without giving credit to these publications. The work by Hatini et al is mentioned only perfunctorily despite it being the most extensive analysis of JAK/STAT function on wing disc development.

1.3- Key stocks like the upd-Gal4 or the UAS-domeDN, UAS-hop-Myc are mentioned under supplementary methods without crediting the publications, while a long and tedious list of Bloomington and Vienna stocks is included in the main text.

The paper by Hatini et al shows that STAT is required sequentially to coordinate the subdivision of the wing disc. The manuscript by Recasaens-Alvares presents similar results but ignoring the previous work. The result will be the existence of two publications that deal on the same topic but don't build on each other. Fig S6 B,E in Hatini et al. is reminiscent of Fig 2F in this manuscript, however no mention to the Hatini paper appears here. Fig 1 in this manuscript is very similar to Fig.1 in Hatini et al.

At the end of page 5 in the Results section the manuscript reads:

"These results indicate that JAK/STAT is required for proper wing fate specification".

It would be fairer to say "These results confirm,..." as Hatini et al already did that claim in their 2013 publication.

The expansion of mirror expression shown in figure 2i and j caused by JAK/STAT RNAi down regulation is to some extent predictable from the Hatini model: STAT blocks proximal notum and it allows the appearance of medial and lateral fates as it normally down-regulates them.

In the manuscript the authors say:

"Interestingly, ectopic expression of Upd to the most proximal side of the wing primordium reduced the expression levels of mirror throughout development (Fig. 3a-c), caused a reduction in the size of the notum (Fig. 3d, compare with the inset in Fig. 3d)".

This has been shown previously by Hatini, et al Fig 3 as they describe that Evg, Wg and Mirror are absent from the notum when Upd is expressed ectopically with Esg-Gal4.

2- The section JAK/STAT restricts the expression of its own ligand is full of indirect results and not very convincing.

2.1- The authors say:

"RNAi-mediated depletion of JAK/STAT in the most proximal part of the wing primordium caused a non-autonomous increase in the levels of the 10xSTAT-GFP activity reporter (Fig. 3h, i, red arrow)".

It was shown by Hatini et al. that stat mutant clones generate ectopic wing discs (similar to the ones the authors of this manuscript induce in the RNAi experiments). The ectopic expression of 10xSTAT GFP observed can be explained by the formation of an ectopic hinge, and not by blocking a putative upd driven negative feed-back loop.

2.2- The other observation suggesting a negative feed-back loop is the induction of ectopic upd-Gal4 >GFP expression in the wing pouch. However the authors do not focus at the relevant place: the notum. As their claim in the Discussion section is that JAK/STAT can bypass EGF repression in the proximal region to generate ectopic wings, it is in the notum and not in the wing where such feed back loop is relevant. In fig 3o despite having induced stat- clones in the notum, the single channel close-up focuses on the wing clones, why? Is no ectopic GFP observed in the notum? Also, I believe it would be more appropriate to test by in situ the ectopic activation of upd RNA. upd-Gal4 is not a direct test of upd expression and may give rise to artefacts.

3- In Fig 5g the levels of the endogenous dIAP1 protein should be shown rather than dIAP1-lacZ that has been selected as a site with STAT binding sites

4- The requirement of Stat92E activity on posterior compartment cells and its relationship to engrailed function is very intriguing and should be studied more in detail. Finding a physiological connection would reinforce this manuscript.

4.1- The most extreme case of anterior vs. posterior area difference occurs in the notum. It would be interesting to analyze carefully if STAT is required there.

4.2- Authors should report if the requirement of STAT in the posterior compartment is specific to the wing or if it is generally required in the posterior compartments of other discs, the leg discs for example.

5- The rescue of JAK/STAT lack of function by enRNAi in Fig 5q and related experiments need controls. Some engrailed enhancers are autoregulated. As en-Gal4 is used for driving UAS-

domeDN, a control for the level of expression is required.

6- There are some unsupported conclusions like:

"Our results indicate that Upd and the JAK/STAT pathway restrict EGFR signaling to the most proximal part of the wing primordium, thereby interfering with the loop and allowing Wg to correctly trigger wing development."

No direct evidence of EGFR regulation is presented.

Similarly:

"Thus, while Wg plays an instructive role in wing fate specification, the Notch and JAK/STAT pathways play a permissive role in this process by restricting the activity range of the antagonizing signalling molecule Vn to the body wall region (Fig. 3n)."

No direct evidence of EGFR regulation is presented.

Or:

"Here we unravel an unprecedented role of the JAK/STAT pathway in maintaining the pool of Hh producing cells by ensuring their cycling and survival. JAK/STAT regulates the expression of *Drosophila* inhibitor of apoptosis 1 (dIAP1) and the G2/M cyclin CycA in P cells, and it does so by counteracting the negative effects of En on these two genes."

Is risky to conclude this. If engrailed was responsible for the STAT phenotypes, the down-regulation of en with RNAi would be expected to have a similar effect on wing size as the down-regulation of JAK/STAT function.

Reviewers' comments:

Reviewer #1 (Remarks to the Author):

The paper by Recanses-Alvarez et al. examines in detail the roles of the JAK/STAT pathway in wing development and identifies three distinct phases. During the 2nd instar, expression of Upd in the proximal side of the disc is required to attenuate EGFR activity, but does not play a role in defining the opposite expression domains of Wg and Vn. During the 3rd instar the JAK/STAT pathway attenuates the negative effects of En on cell division. Thus, JAK/STAT is specifically required in the posterior compartment, for its proper growth. In extreme cases when the P compartment is extremely small, there are secondary effects on growth due to low levels of Hh and low levels of Dpp. Finally, division of cells in the wing hinge, promoted by the JAK/STAT pathway, insulate the hinge from the effects of Dpp in the wing pouch, and allows its proper development.

The experiments are carefully executed, and the conclusions are well justified by experimental data. The precise definition of roles for JAK/STAT during wing development is important, and some attempts to get at the actual mechanism have been made, when relevant target genes that are affected by the JAK/STAT pathway were identified. This paper contributes important aspects to our global understanding of the stepwise processes leading to wing growth and development. Although not particularly inspiring, this paper should be published and will be of interest to a broad audience.

We appreciate Reviewer's comments on the quality of the paper and their conclusions as well as on the interest of the paper to the broad audience. We are convinced this work will inspire further research in the vertebrate limb field.

Reviewer #2 (Remarks to the Author):

Recasaens -Alvares et al present a paper where they discuss the involvement of JAK/STAT signaling in the proliferation and patterning of the wing imaginal disc. The effect of stat92E on posterior compartment cells is very intriguing and is by far the strongest part of the manuscript.

STAT has been previously shown to give competitive growth advantage to cells. The work presented here shows that besides its requirement for cell competition, STAT is required in the posterior compartment of the wing to maintain cell proliferation and viability. The manuscript is interesting but has a major fault likely to irritate many JAK/STAT signaling researchers: it does not credit sufficiently previously published work.

We appreciate Reviewer's comment on the interest of the paper. We have re-organized the introduction and included some sentences throughout the ms to increase the credit to previously published work on the role of the JAK/STAT pathway in the development of Drosophila limbs.

The first 24 lines of the introduction are too general and uninformative. They deal on the importance of morphogens without specifying which ones. It is only when they start talking about hh when they mention relevant data.

We believe that the first paragraph is absolutely necessary to introduce the relevant concepts in the field of morphogens and limb development. As suggested by the reviewer, we have re-organized the second paragraph of the Introduction to specify which morphogens and signaling pathways are involved in the specification of the wing and body wall regions, as they are relevant for the major discoveries of the paper. We appreciate Reviewer's suggestion.

In comparison, the introduction on Jak/Stat, the main topic of the manuscript, is reduced to three lines. Very relevant papers dealing on topics at the heart of this work should be cited, these include Hatini et al 2013 Dev Biol. 378:38-50 and Rodrigues et al. 2012 Development 139:4051-4061. These are relevant papers that, as the manuscript is written, seem hidden away.

We have re-organized the third paragraph of the Introduction to expand the introduction on JAK/STAT and to include the suggested citations on the role of the JAK/STAT pathway in patterning the limb primordia along the proximal-distal axis (Hatini et al, 2013; Ayala-Camargo et al, 2007,

2013) and in regulating growth and the competitive status of proliferating cells (Mukherjee et al, 2005; Rodrigues et al. 2012).

Major points:

1-Not enough credit is given to prior JAK/STAT work. Primary papers that showed very relevant results to the topic of the manuscript are ignored. For example:

1.1- Mukherjee et al Oncogene (2005) 24, 2503-2511, mentions the regulation of CyclinB by the JAK/STAT pathway and describes the small disc phenotype of hop mutant discs. Both topics are introduced in the paper without mentioning the prior work, giving the wrong impression that these observations are new.

We have included the suggested citation on the role of the JAK/STAT pathway in regulating growth in the third paragraph of the introduction and in the Results section as follows:

Introduction (pg 4): "This pathwayregulates growth and the competitive status of the proliferating cells (Rodrigues et al, 2012; Mukherjee et al, 2005).

Results (pg 8): "A similar reduction in the size of the wing disc was observed in *hop*²⁷ mutant animals (Fig. 4d, see also Mukherjee et al, 2005), and in this background the P compartment was also reduced in size (Fig. 4d)."

Results (pg 11): "*In contrast, CycA and B levels were visibly reduced in P cells depleted of Dome activity (Fig. 5m and Supplementary Fig. 3, see also Mukherjee et al, 2005).*"

1.2- Similar experiments to those described at the start of the Results section in Fig1c-d have been published by Hatini et al 2013 (Dev Biol) and by Rodrigues et al Development 2012 and here are reported without giving credit to these publications. The work by Hatini et al is mentioned only perfunctorily despite it being the most extensive analysis of JAK/STAT function on wing disc development.

Reviewer is right in claiming that the groups of V. Hatini and E. Bach have previously analyzed the expression pattern of Upd and the activity of the JAK/STAT pathway during wing development, and references to their papers are already included in pg 8 as follows: "*Expression of Upd evolves as wing development proceeds and the ligand becomes restricted to the presumptive wing hinge - a region that connects the developing wing to the surrounding body wall (Rodrigues et al, 2012; Hatini et al, 2013; Ayala-Camargo et al, 2013; Johnstone et al, 2013)*"

However, the restricted expression of Upd to the distal wing primordium and the graded activity of STAT along the proximal-distal axis in early second instar wing discs that we show in Figure 1 of our manuscript were not reported and/or observed in their publications. Indeed, both papers concluded that the ligand and the pathway were ubiquitously expressed and activated in early wing discs:

- (1) Concerning STAT signaling in early wing discs, Hatini et al 2013 included statements like "We found that at second instar, *stat92E* was active throughout the disc proper of the wing disc" or "We find that *stat92E* is active ubiquitously at early stages". Similarly, Rodrigues et al, 2012 indicated that "*in a mid-second instar disc (~60 hours AED), Stat92E activity is detected at higher and equal levels in nearly all cells, including those in the pouch*".
- (2) With respect to Upd expression in early wing discs, Rodrigues et al, 2012 concluded that "*upd must be expressed early and broadly during development*" and Hatini et al, 2013 that "*We found that most wing disc cells originated from upd-GAL4 expressing cells except for a small cell population near the disc stalk*".

So, we do not believe we should give more credit to their work at this point.

1.3- Key stocks like the upd-Gal4 or the UAS-domeDN, UAS-hop-Myc are mentioned under supplementary methods without crediting the publications, while a long and tedious list of Bloomington and Vienna stocks is included in the main text.

As suggested by the reviewer, we have re-organized the Methods in both the main ms and the supplemental information document as follows: (1) The relevant stocks to modulate or visualize

JAK/STAT activity are in the Methods section of the main ms and the corresponding publications are now included; (2) Description of some stocks have been transferred to the supplemental methods.

Reviewer should note, though, that references to most of these tools were already included in the main text.

The paper by Hatini et al shows that STAT is required sequentially to coordinate the subdivision of the wing disc. The manuscript by Recasaens-Alvares presents similar results but ignoring the previous work. The result will be the existence of two publications that deal on the same topic but don't build on each other.

In our work, we present evidence (by gain and loss of function experiments) that JAK/STAT contributes early in development to Wg-dependent wing fate specification by restricting the expression of EGFR-target genes and notum specification to the proximal part of the primordium.

By contrast, Hatini et al 2013 conclude that JAK/STAT antagonizes wing fate specification as stated in their abstract, at the end of their introduction, in their results section and in their discussion with sentences like: “We also show that the early ubiquitous activity of *stat92E* is required to inhibit ectopic wing induction in ectopic locations”, or “We therefore propose that the broad competence of the early wing disc to respond to wing inducing signals is antagonized in the presumptive notum by *stat92E*”

We believe their conclusion is based on the inappropriate interpretation of the effect of clones of cells mutant for *stat92E*. It is clear from their Figure 2 images that *stat92E* mutant cells do not acquire wing identity cell autonomously, but they do trigger wing development in the adjacent wild type cells, which is consistent with our own data on the non-autonomous induction of wing structures by depletion of JAK/STAT with RNAi (Figure 3k). Indeed, authors concluded in their Results' section that “Ectopic wings were composed of mostly wildtype cells suggesting that following wing induction non-wing cells were recruited to the wing field using a non-autonomous mechanism”. Based on our own data presented in Figure 3k-o, and Supplementary Figure 1, we propose that the non-autonomous induction of wing structures is a consequence of the negative feedback loop of the pathway on the expression levels and pattern restriction of the ligand Upd.

We rather prefer not to discuss this discrepancy in our paper to avoid confusion to the reader. However, we included a reference to Hatini's paper on the non-autonomous induction of wing structures in pg 7 as follows:

“Interestingly, RNAi-mediated depletion of JAK/STAT in the most proximal part of the wing primordium caused ... the non-autonomous induction of wing structures, as monitored by the ectopic expression of Nub (Fig. 3k, see also Hatini et al, 2013)”

Fig S6 B,E in Hatini et al. is reminiscent of Fig 2F in this manuscript, however no mention to the Hatini paper appears here.

Our Figures 1f and S1g, h clearly show that loss of JAK/STAT in the whole wing causes loss of wing specification and duplication of notum structures in the resulting adult flies and this phenotype is reinforced in Figures 1 and 2 by the analysis of molecular markers in the developing wing primordia using exactly the same genetic tools.

By contrast, Hatini et al present evidence in their Figure S6B, E that clones of cells mutant for STAT cause “formation of ectopic wings from the posterior part of the notum (Figure S6B)”, and “differentiation of cuticular protrusions decorated with bristles characteristic of the notum in place of naked cuticle characteristic of the pleura surrounding the hinge (Figure S6E)”. Their clones are not marked and they cannot assess whether the effects of their clones are cell autonomous or non-cell autonomous. Thus, it is difficult to conclude much from Hatini et al's Figure S6. We prefer not to include a reference to this paper at this point as this might lead to confusion.

Fig 1 in this manuscript is very similar to Fig.1 in Hatini et al.

The main goal of Hatini's Figure 1 (as stated in the title and legend of Figure 1 and in the title of the first paragraph) is to demonstrate that "Downregulation of *stat92E* activity in the presumptive notum and pouch coincided with the expansion of these primordia marked with *Eyg* and *Nub*, respectively" and to propose "... roles for *stat92E* in the elaboration of the wing PD axis".

By contrast, the goals of our Figure 1 are (1) to unravel the previously uncharacterized restricted expression of *Upd* to the distal wing primordium and the graded activity of STAT along the proximal-distal axis in early wing discs and (2) to characterize the phenotypic consequences in adult wings and wing primordia of depletion of the JAK/STAT pathway, to finally conclude, as stated in the title of our Figure 1 that "JAK/STAT is required for wing fate specification".

At the end of page 5 in the Results section the manuscript reads:

"These results indicate that JAK/STAT is required for proper wing fate specification".

It would be fairer to say "These results confirm,..." as Hatini et al already did that claim in their 2013 publication.

Hatini et al never claimed in their 2013 paper that JAK/STAT is required for proper wing fate specification since they never showed a failure in the specification of the endogenous wing. They concluded exactly the opposite as stated in their discussion section: "...*stat92E* inhibits the induction of ectopic wing fields in part of the wing disc that normally adopts body wall identity (Fig. 1 and Fig. 2, Sup. Fig. 6B)."

We conclude, based on our Figure 1 and 2, that JAK/STAT prevents the formation of ectopic notum, whereas Hatini et al 2013 concludes, as stated in their abstract or results that "JAK/STAT prevents the formation of ectopic wing fields" or "*stat92E* represses inappropriate induction of ectopic wing fields". We rather prefer not to discuss this discrepancy in our paper, as it is based (as discussed in a previous response to this reviewer) on Hatini's inappropriate interpretation of the effect of STAT mutant clones in the body wall.

The expansion of mirror expression shown in figure 2i and j caused by JAK/STAT RNAi down regulation is to some extent predictable from the Hatini model: STAT blocks proximal notum and it allows the appearance of medial and lateral fates as it normally down-regulates them.

In the manuscript the authors say:

"Interestingly, ectopic expression of Upd to the most proximal side of the wing primordium reduced the expression levels of mirror throughout development (Fig. 3a-c), caused a reduction in the size of the notum (Fig. 3d, compare with the inset in Fig. 3d)".

This has been shown previously by Hatini, et al Fig 3 as they describe that Eyg, Wg and Mirror are absent from the notum when Upd is expressed ectopically with Esg-Gal4.

In Figures 2 and 3, we use a combination of genetic tools to induce loss or ectopic activation of JAK/STAT, and analyze the molecular markers that explain the phenotypes observed in adult flies presented in Figure 1. Interestingly, this analysis is being made not only in late third instar discs but also in second instar stages, the developmental point at which the decision between wing vs notum is taking place. We present evidence that loss of JAK/STAT in the whole primordium gives rise to loss of wing markers (eg. *Nubbin*) and to the distal expansion of notum markers (eg. *mirror*). The resulting adult flies have no wing and their notum is duplicated. Ectopic activation of JAK/STAT in the proximal side of the wing disc results in the downregulation of notum markers, and this is consistent with a reduced notum size and with the induction of ectopic wing structures. We can then conclude that JAK/STAT restricts the specification of the notum to the proximal domain of the wing primordium.

Figure 3H and I of Hatini et al 2013's paper show that broad ectopic expression of *Upd* with the *esg-Gal4* driver induced the loss of *mirror-lacZ* expression (a notum marker), which is consistent with our observations. However, their Figure 5C shows exactly the opposite result. When *upd* is ectopically expressed in the most proximal side of the wing primordium with the *ptc-Gal4* driver, authors claim that *mirror-Z* is expanded. Thus, it is rather difficult to interpret Hatini's data on the effect of *Upd* and JAK/STAT on *mirror* expression. We prefer not to include a reference to this paper at this point as this might lead to confusion.

2- The section JAK/STAT restricts the expression of its own ligand is full of indirect results and not very convincing.

2.1- The authors say:

"RNAi-mediated depletion of JAK/STAT in the most proximal part of the wing primordium caused a non-autonomous increase in the levels of the 10xSTAT-GFP activity reporter (Fig. 3h, i, red arrow)". It was shown by Hatini et al. that *stat* mutant clones generate ectopic wing discs (similar to the ones the authors of this manuscript induce in the RNAi experiments). The ectopic expression of 10xSTAT GFP observed can be explained by the formation of an ectopic hinge, and not by blocking a putative upd driven negative feed-back loop.

As stated above, it is clear in Figure 2 of Hatini's paper that *stat92E* mutant cells do not acquire wing identity cell autonomously, but they do trigger wing development in the adjacent wild type cells, which is consistent with our own data on the non-autonomous induction of wing structures by depletion of JAK/STAT with RNAi (Figure 3). Indeed, Hatini et al concluded in their Results' section that "Ectopic wings were composed of mostly wildtype cells suggesting that following wing induction non-wing cells were recruited to the wing field using a non-autonomous mechanism". Based on our own data presented in Figure 3i-o, and Supplementary Figure 1, we propose that the non-autonomous induction of wing structures is a consequence of the negative feedback loop of the pathway on the expression levels of the ligand Upd and on the restriction of its pattern of expression. Consistent with this feedback loop, and as stated in our ms (pg 7): "... expression of a truncated form of the Domeless receptor (*Dome^{DN}*), which lacks the intracellular domain but is potentially able to trap the ligand, did not phenocopy the non-autonomous effects of RNAi-mediated depletion of JAK/STAT (Fig. 3l), and, remarkably expression of *Dome^{DN}* was able to fully rescue the non-autonomous induction of wing structures caused by targeted expression of *dome^{RNAi}*". We believe that the rescue experiment presented in our Figure 3m clearly shows that the non-autonomous induction of ectopic wing structures is Unpaired-dependent.

2.2- The other observation suggesting a negative feed-back loop is the induction of ectopic *upd-Gal4 >GFP* expression in the wing pouch. However the authors do not focus at the relevant place: the notum. As their claim in the Discussion section is that JAK/STAT can bypass EGF repression in the proximal region to generate ectopic wings, it is in the notum and not in the wing where such feed back loop is relevant. In fig 3o despite having induced *stat*- clones in the notum, the single channel close-up focuses on the wing clones, why? Is no ectopic GFP observed in the notum? Also, I believe it would be more appropriate to test by *in situ* the ectopic activation of *upd* RNA. *upd-Gal4* is not a direct test of *upd* expression and may give rise to artefacts.

We appreciate Reviewer's comment on this issue. We have now analyzed the expression of *upd-RNA* by *in situ* hybridization in wing discs in which the JAK/STAT pathway is depleted in the A compartment. As now shown in new Figure 3o and Figure S1, the expression pattern of *upd* is highly dynamic during development. In second instar wing discs, *upd* is first expressed in the distal part of the wing primordium and soon after, once the wing has been specified, it starts to retract from the pouch. In early third instar wing discs, it gets restricted to the wing hinge and, as development proceeds, *upd* expression is progressively resolved into its characteristic five-spot pattern in the hinge. As shown in new Figure 3o, in JAK/STAT depleted wing discs, the restriction of *upd* expression is impaired and the expression levels are increased. However, and as noted by the Reviewer, *upd* is not ectopically expressed in the notum. These data are consistent with the observation that clones of cells mutant for STAT show ectopic expression of the *upd-Gal4* in the wing but not in the notum (data now included as Supplementary Fig. 1). We have then changed our conclusion as follows in the text: "these observations support the notion that the negative feedback loop between JAK/STAT signalling and its ligand contributes to restrict the expression levels and pattern of Upd to the maturing wing hinge (Fig. 3n) and that a failure to do so interferes with the wing vs body wall subdivision." Since "JAK/STAT depletion did not cause the ectopic expression of *upd* in the body wall of early discs (Fig. 3o and Supplementary Fig. 1)" we also concluded in the text that "...the mechanism by which the early expression of *upd* is restricted to the distal wing primordium remains to be elucidated."

We have also incorporated in Supplementary Fig. 1 an example of an early second instar wing disc to show, by *in situ* hybridization with an *upd* RNA probe, that *upd* expression is restricted to the distal wing primordium, thus validating the use of the *upd-gal4* driver to monitor *upd* expression. This new

data is discussed in the text (pg. 5) as follows: “We confirmed the restricted expression of *upd* to the distal domain of early wing discs by *in situ* hybridization (Supplementary Fig. 1).”

3- In Fig 5g the levels of the endogenous *DIAP1* protein should be shown rather than *DIAP1-lacZ* that has been selected as a site with STAT binding sites

The *DIAP1-lacZ* that we show in Fig5 is a regular *DIAP1* enhancer trap that integrates all inputs on *DIAP1* expression, and whose Bloomington stock number code is 12093 (as indicated in our Methods' section). In our experiments, we have not used the *DIAP1-lacZ* construct with the STAT binding sites generated by Betz et al, 2008. The suggested experiment on the impact of increased JAK/STAT signaling on *DIAP1* protein levels was already performed in Betz et al, 2008. To make it clearer in the text, we have rephrased the sentence in pg 10 as follows: “Consistent with this report, overexpression of *Hop* led to a cell-autonomous increase in the expression levels of a *DIAP1* enhancer-trap (compare Fig. 5e and g).”

4- The requirement of” *Stat92E* activity on posterior compartment cells and its relationship to engrailed function is very intriguing and should be studied more in detail. Finding a physiological connection would reinforce this manuscript.

We have already found a physiological connection between *Stat92E* activity and *Engrailed*. As we conclude in our discussion, “the capacity of *En* to block cell cycle and promote cell death ... is specifically suppressed in the developing wing by JAK/STAT, thus allowing *En*-dependent induction of *Hh* expression and promoting *Dpp*-mediated appendage growth (Figure 7g)”.

There are many genes whose activity is permanently downregulated or reduced during development, as they are specifically used in stress conditions (eg. *p53*, *TNF*) or they have an important role in other developmental context (eg. *apterous* activity is downregulated during late wing development to facilitate the changing pattern of *Serrate* and *Delta*, Milán and Cohen, 1999). The latter is proposed for engrailed in our discussion as follows: “the capacity of *En* to block cell cycle and promote cell death might be required in another developmental context ...”.

Interestingly, *En*-expressing territories in the embryonic ectoderm are highly enriched in apoptotic cells as shown a long time ago by Pazdera et al., 1998. Whether this apoptosis plays a biological role and relies on *En* activity requires further study and we believe it is out of the scope of this work.

4.1- The most extreme case of anterior vs. posterior area difference occurs in the notum. It would be interesting to analyze carefully if STAT is required there.

We have included a new experiment in our ms where we present evidence that targeted activation of the JAK/STAT pathway in the whole posterior compartment (in *en-gal4*, *UAS-hop* wing discs, Supplementary Fig. 1) induces the overgrowth of the P compartment of the notum (the posterior wing pouch is mildly affected) and the induction of ectopic wings. These data are now included in Supplementary Fig. 1 to support the conclusion that activation of JAK/STAT in the body wall induces the appearance of ectopic wing structures. Whether the overgrowth of the P compartment in the notum is just a consequence of the appearance of ectopic wing structures or whether JAK/STAT directly promotes growth of the P compartment of the notum remains to be elucidated. Data are discussed in the text (pg. 7) as follows: “Ectopic wing structures in the body wall were also observed in wing discs as well as in the resulting adults when *Hop* was overexpressed in the posterior compartment (Supplementary Fig. 1)”.

4.2- Authors should report if the requirement of STAT in the posterior compartment is specific to the wing or if it is generally required in the posterior compartments of other discs, the leg discs for example.

As suggested by the Reviewer, we have now included new data to conclude that the requirement of STAT in the posterior compartment is also valid for the haltere and leg imaginal discs. Data are included as Supplementary Fig. 2h-m and discussed in the paper as follows (pg 9): “Interestingly, the size of the P compartment was also reduced in JAK/STAT-depleted haltere and leg primordia (Supplementary Fig. 2)”

5- The rescue of JAK/STAT lack of function by *enRNAi* in Fig 5q and related experiments need controls. Some engrailed enhancers are autoregulated. As *en-Gal4* is used for driving *UAS-domeDN*, a control for the level of expression is required.

We appreciate Reviewer's comment on this issue. As now included in pg 12 of the ms: *"The rescue in tissue size, apoptosis and CycA protein levels caused by expression of en^{RNAi} was not an indirect consequence of a reduction in the expression levels of the en-gal4 driver. If anything, the expression levels of this driver increased, monitored by an UAS-GFP transgene and antibodies to the Gal4 protein (Fig. 6e, f, quantification in Fig. 6g)."*

These new findings are consistent with the observations presented by Garaulet et al, 2008's paper in which authors present evidence that Engrailed over-expression in the wing negatively regulates its own expression. We have now discussed our observations together with the ones presented by Garaulet et al, 2008 in the Discussion as follows:

"The capacity of En to negatively regulate its own expression was subsequently shown to be mediated by the Polycomb-group genes and proposed to be used to finely modulate physiological En expression levels (Garaulet et al, 2008). Consistent with this proposal, we observed an increase in the expression levels of the en-gal4 driver - which is inserted in the en locus and behaves as a transcriptional reporter – in en-RNAi-expressing wing discs (Fig. 6e-g)."

"We propose that the capacity of En to block cell cycle and promote cell death ... is specifically suppressed in the developing Drosophila limbs by JAK/STAT, and modulated by the negative autoregulation of En, thus allowing En-dependent induction of Hh expression and promoting Dpp-mediated appendage growth"

6- There are some unsupported conclusions like:

"Our results indicate that Upd and the JAK/STAT pathway restrict EGFR signaling to the most proximal part of the wing primordium, thereby interfering with the loop and allowing Wg to correctly trigger wing development."

No direct evidence of EGFR regulation is presented.

We have reformatted this sentence as follows: *"Our results indicate that Upd and the JAK/STAT pathway restrict the expression of EGFR target genes to the most proximal part of the wing primordium, thereby interfering with the loop and allowing Wg to correctly trigger wing development."*

Similarly:

"Thus, while Wg plays an instructive role in wing fate specification, the Notch and JAK/STAT pathways play a permissive role in this process by restricting the activity range of the antagonizing signalling molecule Vn to the body wall region (Fig. 3n)."

No direct evidence of EGFR regulation is presented.

We present evidence in this work that JAK/STAT signaling has a negative impact on the expression of Vein/EGFR target genes mirror and apterous (and Vein). So, it is fair to conclude that the *"JAK/STAT pathway plays a permissive role in this process by restricting the activity range of the antagonizing signalling molecule Vn to the body wall region."* We are not claiming that JAK/STAT regulates EGFR signaling directly.

Or:

"Here we unravel an unprecedented role of the JAK/STAT pathway in maintaining the pool of Hh producing cells by ensuring their cycling and survival. JAK/STAT regulates the expression of Drosophila inhibitor of apoptosis 1 (dIAP1) and the G2/M cyclin CycA in P cells, and it does so by counteracting the negative effects of En on these two genes."

Is risky to conclude this. If engrailed was responsible for the STAT phenotypes, the down-regulation of en with RNAi would be expected to have a similar effect on wing size as the down-regulation of JAK/STAT function.

We appreciate Reviewer's comments. We have now included in Figure 6k, l and quantified in Figure 6m the suggested experiment where ectopic expression of Engrailed has a similar effect on the size of the posterior compartment as the down-regulation of JAK/STAT function. These new results reinforce our conclusion that JAK/STAT maintains the pool of Hh producing cells by counteracting the negative effects of En.

Reviewer #2 (Remarks to the Author)

In their reviewed manuscript the authors have dealt with most of my criticisms and suggestions.

There are some minor issues pending:

Please clarify in legend that the ectopic wing in Sup Fig1j is made from the mesothorax and not from the metathorax

In page 7 section "jak/stat restricts the expression of its own ligand" it says:

We observed that RNAi depletion of JAK/STAT...Please, specify if RNAi treatment tested was for dome, Stat, hop, for all of them or only dome and stat.

In this experiment, can you explain why the ectopic wings were only induced non autonomously?

What is the evidence for the RNAi specificity? Do dome RNAi and STAT RNAi give similar phenotypes?

REVIEWERS' COMMENTS:

Reviewer #2 (Remarks to the Author):

In their reviewed manuscript the authors have dealt with most of my criticisms and suggestions.

There are some minor issues pending:

Please clarify in legend that the ectopic wing in Sup Fig1j is made from the mesothorax and not from the metathorax

We appreciate Reviewer's suggestion to include the following sentence that clarifies the origin of the ectopic wing in Sup Fig1j: "Red arrows in j point to ectopic wings arising from the mesothorax".

In page 7 section "jak/stat restricts the expression of its own ligand" it says:

We observed that RNAi depletion of JAK/STAT...Please, specify if RNAi treatment tested was for dome, Stat, hop, for all of them or only dome and stat.

We have changed the sentence as follows:

"We observed that RNAi-mediated depletion of Domeless or STAT92E in the anterior compartment of the wing primordium caused a non-autonomous increase in the levels of the 10xSTAT-GFP activity reporter (Fig. 3h, i, red arrow),...."

In this experiment, can you explain why the ectopic wings were only induced non-autonomously?

Our data indicate that the ectopic wings induced in a non-autonomous manner by Dome^{RNAi} are Upd-dependent, as the levels of Upd are higher under these circumstances (Figure 3o) and co-expression of Dome^{DN} rescued this effect (Figure 3k, m). In *ci-gal4, UAS-Dome^{RNAi}* wing discs, the A compartment cannot respond to Upd (as expression of its receptor is being depleted) and the levels of the EGFR target genes promoting notum fate specification are expected to be high, thus inhibiting wing fate specification. By contrast, in the nearby P compartment, increased JAK/STAT activity (as a consequence of Upd coming from A cells) results in EGFR target gene downregulation and to the induction of ectopic wings. It is interesting to note in this context that the posterior notum appears to be a region more competent to trigger wing development. In fact, overexpression of Wg, which can induce Nub expression anywhere in the wing disc, triggers wing development specifically in the posterior notum (Ng et al, 1996).

What is the evidence for the RNAi specificity? Do dome RNAi and STAT RNAi give similar phenotypes?

Both RNAis induce a non-autonomous effect on STAT signaling, mirror expression and wing induction. For simplicity, we have included only one example of each in Figure 3.